# Collective relational inference for learning heterogeneous interactions

Zhichao Han[1], Olga Fink [2,3] & David S. Kammer [1,3] ✉

Interacting systems are ubiquitous in nature and engineering, ranging from particle dynamics in physics to functionally connected brain regions. Revealing interaction laws is of fundamental importance but also particularly challenging due to underlying configurational complexities. These challenges become exacerbated for heterogeneous systems that are prevalent in reality, where multiple interaction types coexist simultaneously and relational inference is required. Here, we propose a probabilistic method for relational inference, which possesses two distinctive characteristics compared to existing methods. First, it infers the interaction types of different edges *collectively* by explicitly encoding the correlation among incoming interactions with a joint distribution, and second, it allows handling systems with variable topological structure over time. We evaluate the proposed methodology across several benchmark datasets and demonstrate that it outperforms existing methods in accurately inferring interaction types. The developed methodology constitutes a key element for understanding interacting systems and may find application in graph structure learning.

Interacting systems that contain a set of interactive entities are omnipresent in nature and engineering. Examples include chemical molecules[1], granular materials[2], brain regions[3] and numerous others[4–6]. The interactive entities are typically represented as a graph where edges correspond to the interactions. As the dynamics of each individual entity and the system behavior arise from interactions between the entities, revealing these interactions and their governing laws is key to understand, model and predict the behavior of such systems. However, the ground-truth information about underlying interactions often remains unknown, and only the states of entities over time are directly accessible. Therefore, determining the interactions between entities poses significant challenges.

The complexity of these challenges increases significantly for heterogeneous systems, where different types of interactions coexist among different entities. An approach that can simultaneously reveal the hidden interaction types between any two entities and infer the unknown interaction law governing each interaction type constitutes a particularly difficult but relevant task. Applications in which relational inference is important are numerous and include, among others, the

discovery of physical laws governing particle interactions in heterogeneous systems[7], unveiling functional connections between brain regions[8], and graph structure learning[9].

Various attempts to this problem have been made in recent years. This includes the neural relational inference (NRI) model proposed by[7], which is built on the variational autoencoder (VAE)[10], and has shown promising results in inferring heterogeneous interactions[8]. However, NRI inherits the assumption of VAE that input data are independent and identically distributed, and, therefore, infers the interaction types for different pairs of entities *independently*. The approach neglects the correlation among interactions on different edges. As the observed states of each entity are the consequence of the cumulative impact of all incoming interactions, conjecturing the interaction type of one edge should take into consideration the estimation of other relevant edges. Neglecting this aspect can result in a significant under-performance, as indicated by[11] and our experiments presented in "Result".

Chen et al.[12] enhanced NRI by including a relation interaction module that accounts for the correlation among interactions.

[1]Institute for Building Materials, ETH Zürich, Laura-Hezner-Weg 7, 8093 Zürich, Switzerland. [2]Laboratory of Intelligent Maintenance and Operations Systems, EPFL, Station 18, 1015 Lausanne, Switzerland. [3]These authors jointly supervised this work: Olga Fink, David S. Kammer. ✉e-mail: dkammer@ethz.ch

**Fig. 1 | Comparison between existing approaches and our proposed approach.**
**a** Neural relational inference (NRI) approach[7] and **b** our proposed method CRI for relational inference. NRI predicts the interaction type of different edges *independently* (e.g., the incoming edges of $v_1$). CRI takes the subgraph of each node (e.g., $S_{(1)}$) as an entity. We learn the joint distribution of the type for all edges in the subgraph, allowing for modeling their *collective* influence on node states. The red bars depict a categorical distribution where the length represents the probability of a particular realization. $\mathcal{F}_v$ and $\mathcal{F}_e$ represent the function approximation of the node state update function and interaction function (by neural networks), respectively. Other mathematical symbols are explained in "Methods" and summarized in the table in Supplementary Information Sec. 1.

Additionally, the study integrated prior constraints, such as symmetry, into the learnt interactions. However, as our experiments will demonstrate, these additional mechanisms prove inadequate in accurately inferring interaction types. Other methods include modular meta-learning[13], as proposed by ref. 14. This approach alternates between the simulated annealing step to update the predicted interaction type of every edge and the optimizing step to learn the interaction function for every type. However, the computation is very expensive due to the immense search space involved, which scales with $\mathcal{O}(K^{|E|})$ for an interacting system containing $K$ different interactions and $|E|$ pairs of interacting entities. Therefore, ref. 14 used the same encoder as NRI[7] to initially infer a proposal distribution for the interaction type of each edge. Subsequently, they used the simulated annealing optimization algorithm to sample possible configurations of the interaction type across different edges. The correlation among interactions on different edges is implicitly captured through this optimization process.

An additional limitation of these existing relational inference methods[7,12,14] is that they are designed to infer heterogeneous interactions in systems with time-invariant neighborhood networks, i.e., systems in which each entity consistently interacts with the same neighbors. In physical systems, it is typical for the network structure of interactions to undergo changes over time as a result of rearrangements. As we will demonstrate, current methods encounter difficulties in effectively learning systems that have an evolving graph topology.

Here, we develop a novel probabilistic approach to learn heterogeneous interactions based on the generalized expectation-maximization (EM) algorithm[15]. The proposed method named *Collective Relational Inference* (CRI) overcomes the above-mentioned challenges. It infers the type of pairwise interactions by considering the correlation among different edges. We test the proposed method on causality discovery and interacting particle systems, which are two representative examples for heterogeneous interaction inference. We demonstrate that the proposed framework is highly flexible, as it allows the integration of any compatible inference method and/or known constraints about the system. This is important, for instance, for learning *physics-consistent* interaction law for granular matter. Further, we propose an extension of CRI, the Evolving-CRI, which is designed to address the challenge of relational inference with evolving graph topology. We empirically demonstrate that CRI outperforms baselines notably in both discovering causal relations between entities and learning heterogeneous interactions. The experiments highlight CRI's exceptional generalization ability, data efficiency, and the ability to learn multiple, distinct interactions. Furthermore, the proposed variant, Evolving-CRI, proves effective in complex scenarios where the underlying graph topology changes over time—a challenge for previous methods. These findings underscore the advantages and necessity of *collective* inference for relational inference.

## Fundamental concept behind the proposed methodology

We use a directed graph to represent an interacting system, where nodes correspond to entities and edges represent their interactions. Our model named CRI is a novel probabilistic approach designed to infer the interaction types of different edges *collectively*. We note that CRI is not the first work aiming to perform CRI. As introduced in "Introduction", ref. 14 aim to implicitly capture the correlation among interactions through simulated annealing optimization, while ref. 12 seek to capture this correlation through their proposed relational interaction neural network. However, CRI fundamentally differs in its approach to capturing the correlations among different edges. Unlike previous methods[12,14], CRI contains the explicit probabilistic encoding of the correlation among different edges, as illustrated in Fig. 1. Specifically, CRI takes into account *subgraphs* comprising a center node and its neighboring nodes as a collective entity and infers the joint distribution of interaction types of the edges within each subgraph *collectively*. The underlying idea behind this approach is that different interactions affect the states of entities collectively. The subgraph representation and the joint distribution explicitly model the *collective* influence from neighbors.

In general, CRI is designed for relational inference for fixed underlying graph topology, and comprises two modules (as depicted in Fig. 2): (1) a probabilistic relational inference module that infers the joint distribution of interaction types of edges in each subgraph, and (2) a generative module that is a graph neural network[16] approximating pairwise interactions to predict new states. The entire model is trained to predict the states of entities over time based on the generalized EM algorithm. This involves iteratively updating the inferred interaction types of edges through the inference module, alongside updating the learnable parameters of the generative module. The details of the CRI methodology are presented in "Collective Relational Inference".

Furthermore, we extend our proposed CRI methodology to address the challenge of relational inference in systems with evolving graph topology, where entities may interact with different neighbors at different times. We introduce a novel algorithm called Evolving-CRI, by adapting the probabilistic relational inference module in Fig. 2. Evolving-CRI is based on the fundamental concept of updating the posterior distribution of possible interaction types for a newly appearing edge. This is achieved by marginalizing out the posterior distribution of all correlated edges. As a result, the interaction type inferred for each edge captures the correlation with other incoming edges, which collectively influence the observed states. The details of

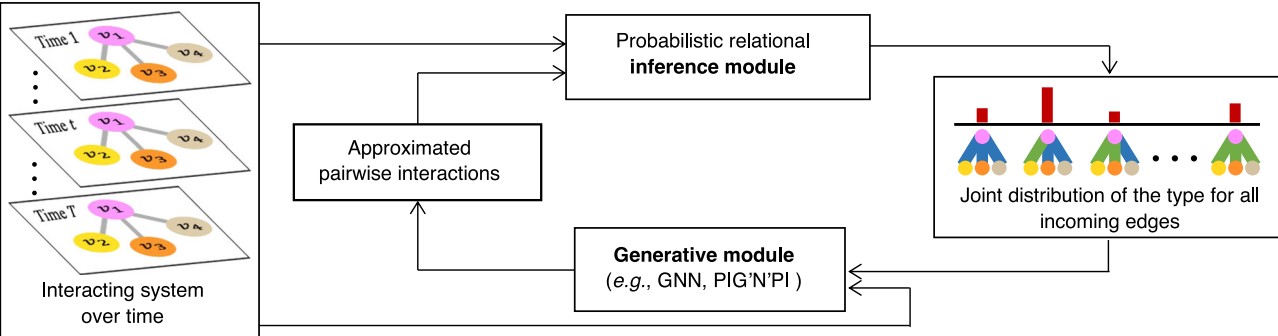

**Fig. 2 | The pipeline of CRI.** The probabilistic relational inference module takes the observed states of the interacting system at different time steps and the current estimation of the pairwise interactions as input, and infers the joint distribution of the interaction type of all edges in each subgraph. The generative module takes the predicted joint distribution for each subgraph together with the observations as input and updates the estimation of different interaction functions. It can be any kind of graph neural network, e.g., the standard message-passing GNN used in[7] or the physics-induced graph neural network[19]. The red bars depict a categorical distribution where the length represents the probability of a particular realization.

Evolving-CRI can be found in "Evolving Collective Relational Inference".

**Design of the inference module.** The inference module is designed for *collective inference* of the interaction types. The desired property of collective inference is achieved by introducing a joint distribution of the types of correlated interactions, based on the Bayesian rules. This explicitly encodes the correlation among different incoming interactions. Such a characteristic distinctly sets CRI apart from previous methods of relational inference as introduced in[7,12,14,17]. We note that the inference module is highly flexible, allowing for the integration of any compatible inference method. For instance, we can substitute the proposed inference module, which infers the exact posterior distribution, with a variational inference method (e.g.,[18]) that infers the approximate distribution. The details of the assessment of this flexibility are presented in Supplementary Information Sec. 2.1. The design of the inference module in Evolving-CRI is adapted from CRI, in which the estimated type of emerging interactions is updated at each new time step. This adaptation alleviates the constraint of fixed neighbors for each entity over time.

**Design of the generative module.** The generative module of CRI can be any type of graph neural networks, including the basic message-passing graph neural networks (see "Relational inference for causality discovery"). Nevertheless, if there are known constraints about the interacting systems, they can also be integrated. For example, the generative module applied for interacting particle systems may ensure that the learnt interactions are *physics-consistent*, meaning they adhere to Newton's laws of motion. In those cases, as described in "Relational inference with known constraints about the interacting system", we use the recently proposed physics-induced graph network for particle interaction (PIG'N'PI)[19].

## Results

To showcase the versatility and performance of the proposed CRI, we evaluate it across three challenging examples: causality discovery, heterogeneous inter-particle interactions inference, and relational inference in a crystallization system with an evolving topology. We compare it to previous methods that infer different edges independently. We consider NRI[7] as the main reference which is the baseline method.

### Relational inference for causality discovery

Causality discovery aims to infer which entities exhibit a causality relationship (see Fig. 3a). The presence or absence of this causality relationship can be considered as two different types of interactions.

Therefore, causality discovery can be seen as a special case of graph structure learning[9] to which relational inference methods can be applied. We acknowledge the existence of other algorithms for causality discovery, such as those based on non-parametric or parametric statistical tests[20,21] and approaches utilizing deep learning[22,23]. Our aim is to assess whether the proposed collective inference offers advantages in relational inference. Therefore, we restrict the comparison to NRI, a representative method inferring relations independently. In this context, we conduct experiments using two benchmark datasets for causality discovery: VAR[11] and Netsim[3,8]. The performance of CRI and NRI is summarized in Fig. 3 and comprehensive results are provided in Supplementary Information Sec. 3.3.

The first considered benchmark comprises vector autoregression (VAR), which is a statistical model used to describe the relationship between multiple quantities, commonly used in economics and natural sciences. VAR time series are created by applying the VAR model with various underlying causality structures (see Fig. 3b). As indicated by ref. 11, the two cases VAR-a and VAR-c present significant challenges for NRI. We follow the procedure outlined in ref. 11 to prepare the training and testing data, ensuring a fair comparison.

The second case study is a realistic simulated fMRI time-series dataset named Netsim. The objective is to infer the directed connections, which represent causal relations among different brain regions. Following the approach outlined in ref. 8, we use data consisting of samples from 50 brain subjects. Each sample comprises $N = 15$ different regions, with each region containing a time series of length $T = 200$. For our process, we allocate the first 30 samples for training, the subsequent 10 samples for validation and the final 10 samples for testing.

In the causality discovery experiment, we use the same standard graph neural network as employed in NRI[7] for CRI, as we lack prior information about the interactions. Viewing the presence and absence of the causality relation as distinct interaction types, we assess the accuracy of interaction type prediction as our evaluation metric.

Our results demonstrate that CRI outperforms NRI significantly in the case of VAR data (see Fig. 3c). Even in challenging instances such as VAR-a and VAR-c, where NRI faces considerable difficulties, CRI demonstrates exceptional performance. These observations align with findings highlighted by ref. 11, who identified NRI's challenges in these cases stemming from the absence of correlations between different edges. As detailed in "Fundamental concept behind the proposed methodology", CRI effectively captures such correlations, which leads to the observed improvement in performance.

Even in the more complex case of Netsim, which has a large number of entities and more correlations between these entities, our results reveal that CRI outperforms NRI for this task (see Fig. 3c). While

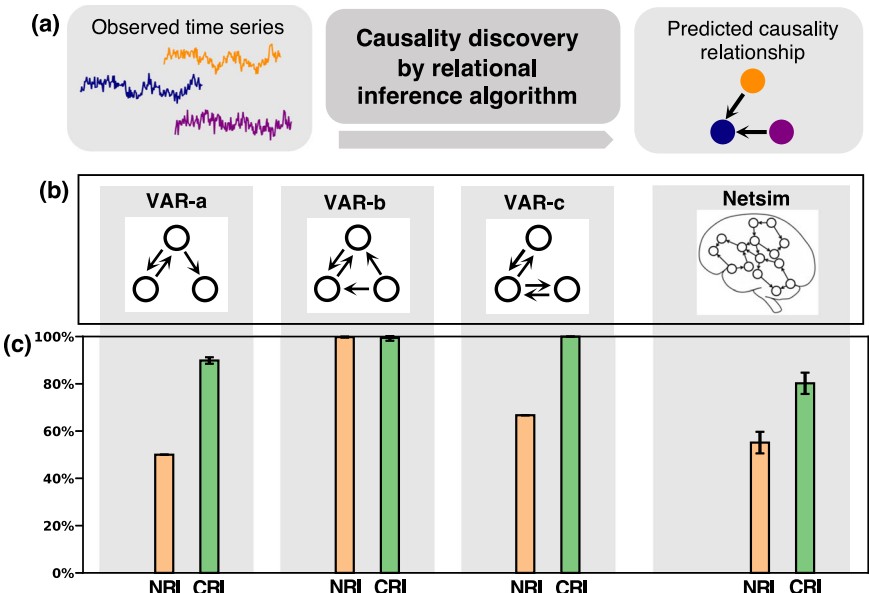

**Fig. 3 | Illustration and results for causality discovery. a** Causality discovery task outline. The state time series of different entities are observed, but whether a pair of entities has a causality relationship is unknown. The relational inference method is expected to infer the correct directed graph structure that represents the underlying causality relationships. **b** The ground-truth causality graph and **c** prediction accuracy for different datasets. Mean and standard derivation are computed from five independent experiments. For NRI on the VAR-a and VAR-c, we report the best performance among the five experiments because some random seeds lead to severe sub-optimal performance for NRI on these two datasets.

it might seem that the Netsim experiment is beyond the capabilities of both CRI or NRI due to the assumption that the underlying dynamics of each interaction type are the same across different edges—something not necessarily true for interactions between different brain areas—CRI demonstrates promising performance. This success is attributed to the collective inference approach.

### Relational inference with known constraints about the interacting system

In certain scenarios, constraints regarding the interacting system are known. For instance, in the case of interacting particle systems, it is crucial that the interactions and the dynamics of entities adhere to Newton's law of motion[19]. To explore this, we consider heterogeneous interacting particle systems governed by Newtonian dynamics, where different types of particles interact via distinct types of forces. We aim to determine whether the proposed CRI can effectively discern these heterogeneous interactions by observing particle trajectories. The task involves a twofold objective: inferring the interaction type between any two particles and learning the associated interactions, which should be *physics-consistent*.

We adapt the simulations of a variety of heterogeneous particle systems, as used in previous studies[7,12], as benchmarks. Specific simulation details are outlined in "Details of the considered case studies". We use PIG'N'PI[19] as the generative module for CRI, ensuring that the learnt interactions are *physics-consistent*. Our comparison involves assessing CRI against NRI[7] and MPM[12], using evaluation metrics defined in "Performance evaluation metrics". Additionally, adaptations are made to both NRI and MPM by substituting their original decoders with PIG'N'PI, denoted by NRI-PIG'N'PI and MPM-PIG'N'PI, respectively. Detailed setups of these baselines are provided in "Configurations of baseline models". Results are summarized in Fig. 4 (comprehensive results are provided in Supplementary Information Sec. 3.4–3.5, 3.7–3.8). Finally, we note that ModularMeta[14] is not included as a baseline in this benchmark due to technical reasons, in particular its slow performance owing to the inner inference. Nevertheless, we have evaluated CRI on the original dataset provided by ref. 7, where CRI demonstrated superior performance on the Charged dataset and

matched the performance on the Springs dataset with a 99.9% accuracy, as detailed in Supplementary Information Sec. 3.2.

First, we evaluate the performance of the proposed methodology in correctly predicting the interaction type. The results demonstrate that CRI significantly outperforms the baselines in classifying the type of each interaction (see Fig. 4-left). The performance gap is more significant for larger systems (e.g., Spring N10K2 vs Spring N5K2, where the former contains 10 particles and the latter has five particles), for systems with more types of interactions (e.g., Spring N5K4 vs Spring N5K2, where the former and latter contain four and two different interaction types, respectively), and complex interaction functions (e.g., complex Charge N5K2 vs simple Spring N5K2). Further, CRI performs much better when training data is limited (e.g., the accuracy of different methods using 100 or 500 simulations for training). Finally, we find that CRI generalizes well with an accuracy greater than 99%, while the accuracy of the best baselines is only about 70% (see Generalization Accuracy in Fig. 4 and Supplementary Information Sec. 3.6). For this evaluation, we used Spring N5K2 as the training and validation dataset, and evaluated the best-performing model (selected using the validation set of Spring N5K2) on the test set of Spring N10K2.

Next, we evaluate the consistency of the inferred pairwise forces with the actual pairwise forces in terms of $MAE_{ef}$ (see definition in "Performance evaluation metrics"). This evaluation is only possible for CRI, NRI-PIG'N'PI and MPM-PIG'N'PI because the original NRI and MPM algorithms learn a high dimensional embedding of the pairwise forces, for which the $MAE_{ef}$ cannot be directly computed. We find that CRI achieves a lower error in learning the pairwise force functions in all systems (see Fig. 4 middle and Supplementary Information Sec. 3.13), particularly for those with limited training data (see the x-axis from 100 to 10 k which is the training data size) and complex scenarios (e.g., simple Spring N5K4 and complex Charge N5K2). We note that NRI-PIG'N'PI and MPM-PIG'N'PI only achieve about 50% accuracy in the Charge N5K2 dataset when the training data is less than 1000, which is similar to random guessing. This implies that their generative module fails to learn any information about the actual force in this range since the generative module depends on the predicted interaction type.

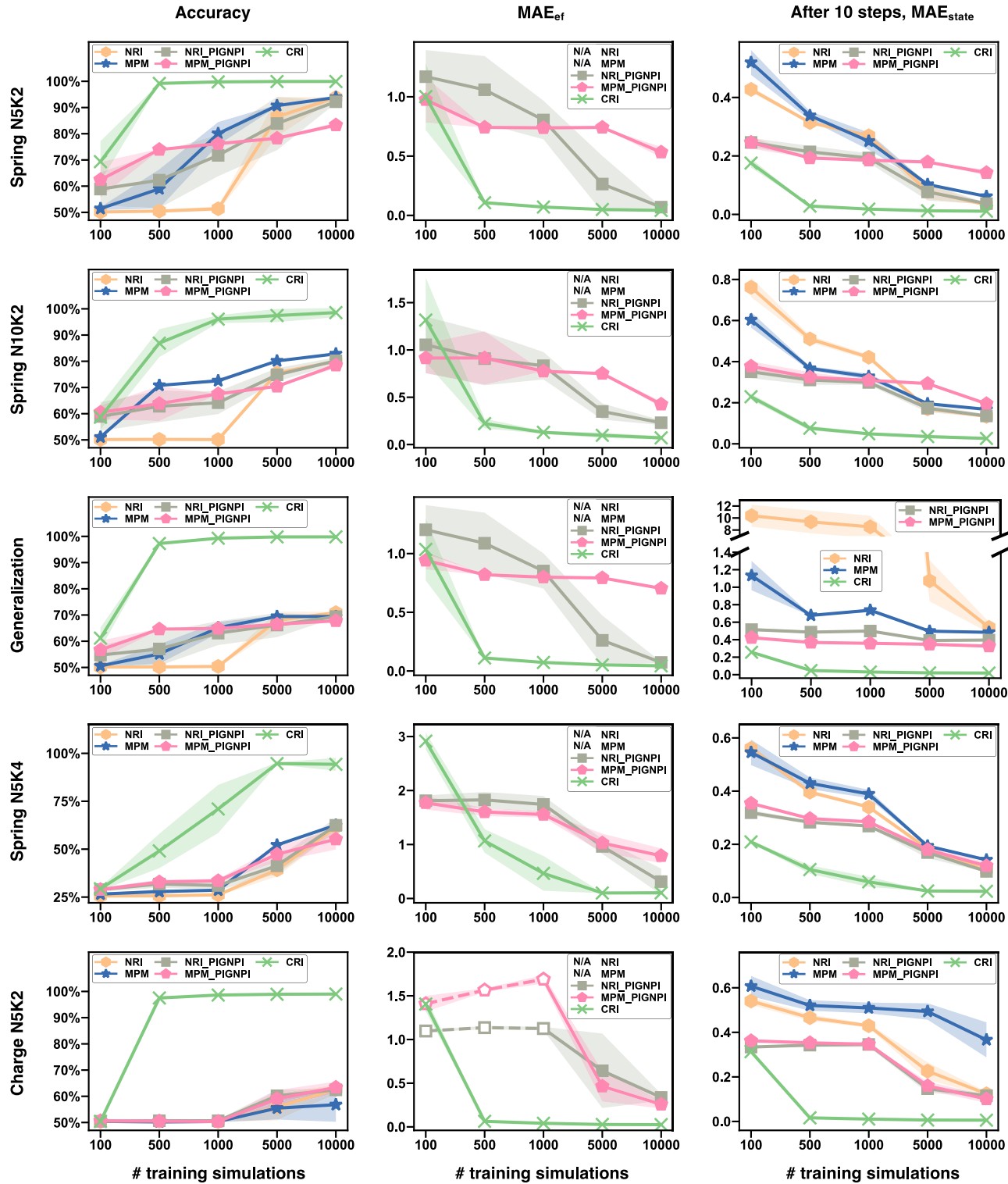

**Fig. 4 | Test performances for the spring and charge experiments.** Mean and standard derivation are computed from five independent experiments. (left column) Accuracy of the interaction type inference. (center column) MAE of pairwise force. NRI and MPM cannot infer pairwise force. Empty symbols with dashed lines in E2 (Charge N5K2) indicate the range in which NRI-PIG'N'PI and MPM-PIG'N'PI do not learn any useful information about pairwise forces. (right column) MAE of state (position and velocity combined) after 10 simulation steps.

Analyzing their learned forces makes no sense with such a limited amount of training data, as indicated by the empty symbols and dashed lines in Fig. 4 Charge N5K2 MAE$_{ef}$. The superior performance of CRI in inferring the interactions, even with limited data access, forms the basis for various downstream applications, such as discovering the explicit form of the governing equations. In such cases, symbolic regression (e.g.,[24]) is applied to search for the best-fitting symbolic expression based on the predicted pairwise force by CRI, as demonstrated in Supplementary Information Sec. 3.14.

Lastly, we evaluate the supervised learning performance of the predicted states (position and velocity) after 10 time steps (see Fig. 4-right). Note that the predicted state, measured by MAE$_{state}$, depends

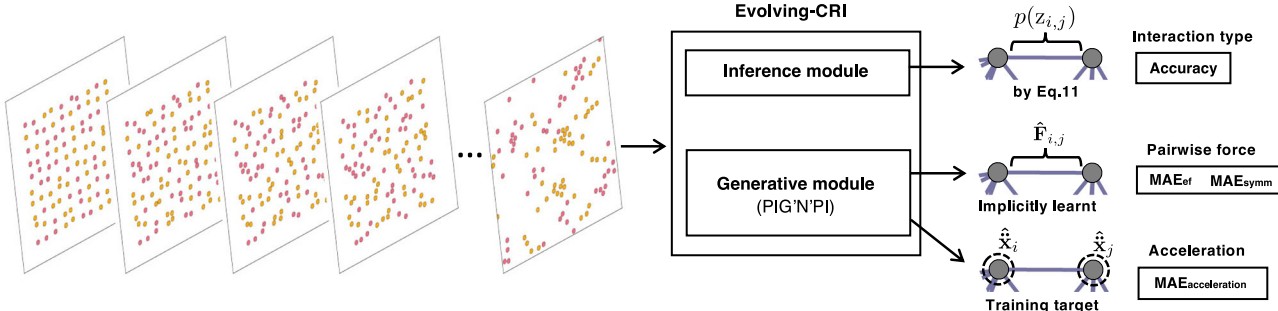

**Fig. 5 | Concept of Evolving-CRI to learn the heterogeneous interactions in crystallization problems.** (left) System evolution during crystallization. Yellow and red colors indicate two different kinds of particles with heterogeneous interactions. (right) Schematic of Evolving-CRI consisting of an inference module and a generative module. Evolving-CRI is trained to predict the ground-truth acceleration. After training, the heterogeneous interactions are implicitly learnt.

on both the accuracy of predicting the type of each interaction and the quality of learning the interaction functions. The results show that CRI achieves much smaller $MAE_{state}$ in all cases.

It is important to note that the good performance of CRI is primarily due to the proposed probabilistic inference method, rather than PIG'N'PI (see ablation study with CRI using a standard graph neural network in Supplementary Information Sec. 3.10). Additionally, we verify and demonstrate that CRI is robust against significant levels of noise (see Supplementary Information Sec. 3.11).

### Relational inference with evolving graph topology

Real-world systems often exhibit more complexity than the benchmark problems considered in previous sections. Many physical systems consist of a large number of entities, and possess interactions that are restricted to a specific neighborhood defined by a critical distance, resulting in a changing topology of the underlying interaction graph over time. To evaluate the ability of Evolving-CRI to handle such complex systems, we examine simulations, adapted from ref. 25. These simulations model the crystallization behavior observed when two different types of particles (e.g., water and oil) are combined (see Fig. 5-left). The system consists of 100 particles with identical mass. Particle interactions are governed by the Lennard-Jones (LJ) and dipole-dipole potentials. Proximity triggers the LJ potential, while the dipole-dipole interaction is attractive for identical particles and repulsive for non-identical ones. These conditions cause particles to reorganize over time, leading to the eventual formation of crystalline structures (see Fig. 5-left).

We evaluate the interpolation ability of each model by randomly splitting the time steps of the entire simulation into training, validation and testing parts. Additionally, we evaluate the models' extrapolation ability by using the first part of the entire simulation for training and validation, and the remaining time steps for testing (details are provided in "Details of the considered case studies"). The model receives particle positions and velocities as inputs and aims to predict accelerations as the target. Due to the evolving topology of the interaction graph over time, adjustments to the baseline models are necessary, as described in detail in "Configurations of baseline models".

The results for both interpolation and extrapolation assessments clearly demonstrate that Evolving-CRI significantly outperforms all considered baselines (see Fig. 6 and Supplementary Information Sec. 3.9). Notably, Evolving-CRI demonstrates accurate prediction of edge types (see Accuracy in Fig. 6a, b), learns the *physics-consistent* heterogeneous interactions without direct supervision (see $MAE_{ef}$ in Fig. 6a, b) and predicts particle states in the subsequent time step (see $MAE_{acceleration}$ in Fig. 6a, b). This stands in contrast to the baselines, which consistently struggle to learn heterogeneous interactions in the particle system with evolving graph topology.

## Discussion

We proposed a novel probabilistic method for relational inference that operates collectively and demonstrated its unique ability to collectively infer heterogeneous interactions. This approach distinguishes itself from and addresses the limitations of previous methods of relational inference. Our initial application of CRI in causality discovery notably outperformed the baseline, NRI, showcasing the advantage of *collective* inference. This success hints at CRI's potential in broader graph structure learning tasks aimed at inferring unknown graph connectivity topologies[9,26,27].

Subsequently, we tested CRI on heterogeneous particle systems where integration of known constraints was feasible in the generative module. CRI demonstrated substantial improvements over baseline models, exhibiting an exceptional generalization ability and data efficiency. Most notably, these experiments highlighted CRI's capability in inferring multiple (>2) interaction types, an area where current state-of-the-art methods struggle. These results hint at the possible application of CRI for heterogeneous graph structure learning[28]. Lastly, we evaluated the performance of the Evolving-CRI variant on complex scenarios where the underlying graph topology evolves over time. Our results showcased its ability to infer heterogeneous interactions effectively, unlike baseline models, which encountered significant challenges.

In summary, our experiments highlight the effectiveness and versatility of the proposed framework, which demonstrates high adaptability and seamless integration with compatible approximation inference methods to infer the joint probability of edge types. The experimental results presented emphasize CRI's potential in enhancing relational inference across diverse applications, including graph structure learning and the discovery of governing physical laws in heterogeneous physical systems.

## Methods

### Graph representation of interacting systems

We explore heterogeneous interacting systems in which entities interact with each other. As mentioned previously, entities can be different correlated regions of the human brain or interacting particles. Each entity undergoes changes in its internal state over time due to these interactions. In such systems, we observe the states of entities at different points in time, lacking specific information about their underlying interactions. These observations can manifest as time-series of signals or the movements of physical particles. Similar to the approach outlined in ref. 7, we assume knowledge of the number of distinct interactions, denoted by $K$. The goal of relational inference in this context is twofold: first, inferring the interaction type between any pair of entities, and second, learning the interaction functions associated with these $K$ different types of interactions. The used mathematical symbols are summarized in the table in Supplementary Information Sec. 1.

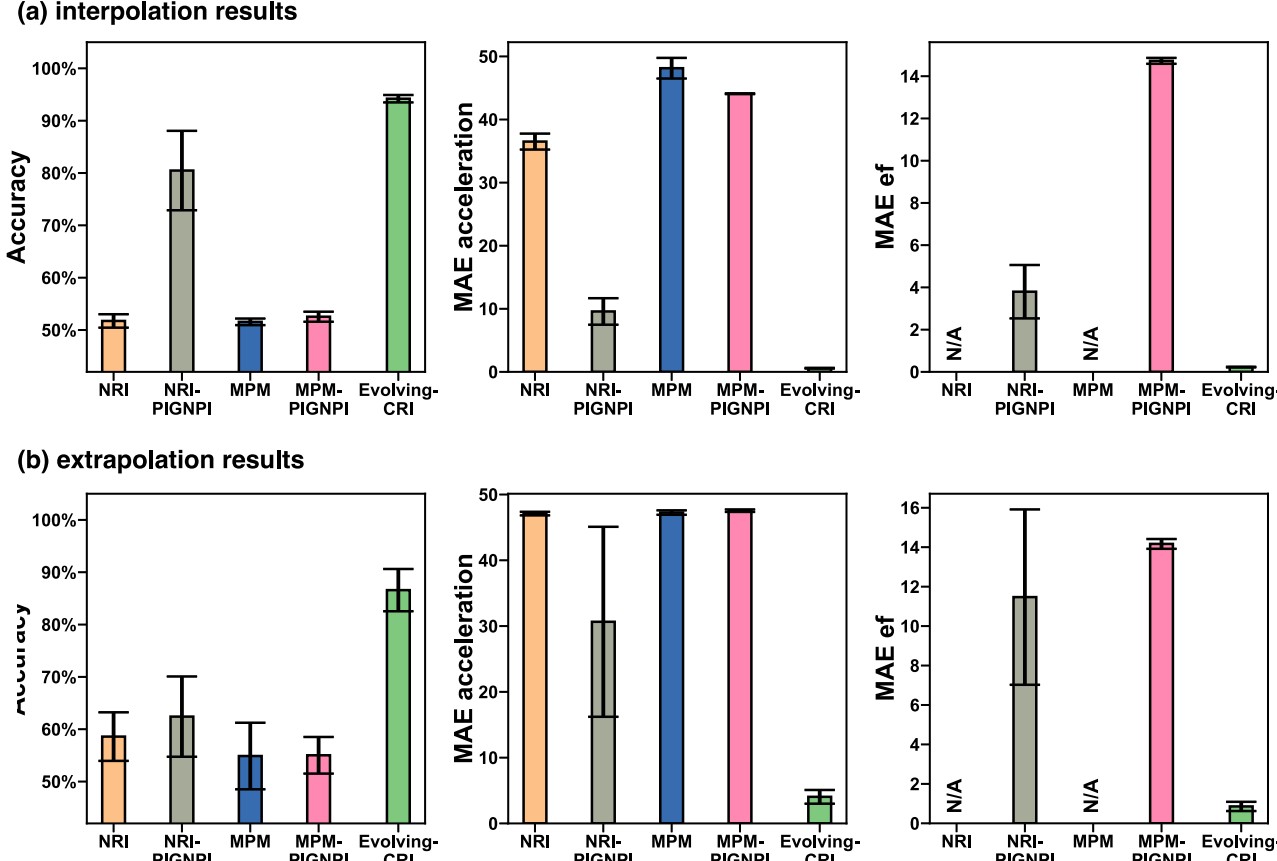

**Fig. 6 | Performances of Evolving-CRI on the crystallization problem with an evolving graph topology. a** Interpolation and **b** extrapolation results of Evolving-CRI and others. Mean and standard derivation are computed from five independent experiments. (left column of **a**, **b**) Accuracy in inferring the interaction type. (center column of **a**, **b**) Mean Absolute Error in particle acceleration. (right column of **a**, **b**) Mean Absolute Error in pairwise interaction. NRI and MPM cannot explicitly predict the pairwise force.

We model the interacting system as a directed graph $G = (V, E)$, where nodes $V = \{v_1, v_2, ..., v_{|V|}\}$ represent the entities and the directed edges $E = \{e_{i,j}|v_j$ acts on $v_i\}$ represent the interactions. Each node $v_i$ is associated with the feature vector $\mathbf{x}_i^t$ that describes the state of $v_i$ at time step $t$. In the causality discovery experiments, the feature of each node is a multivariate time series. In the particle systems, the feature of a node $\mathbf{x}_i^t = [\mathbf{r}_i^t, \dot{\mathbf{r}}_i^t, m_i]$ contains its position $\mathbf{r}_i^t$, velocity $\dot{\mathbf{r}}_i^t$ and mass $m_i$. We use $\Gamma(i) = \{v_j|e_{i,j} \in E\}$ to denote the neighbors having an interaction with $v_i$. Here, we consider two different cases: First, the graph topology remains fixed during the entire time, i.e., the neighbors of each node do not change over time. Second, the underlying graph $G$ has an evolving topology in which nodes interact with different neighbors at different times. In practice, the latter can model realistic physical systems where each node interacts only with nearby nodes, which are within some cutoff radius. In this work, we assume the ground-truth cutoff radius is known, which allows us to focus on developing the relational inference method and to ensure that the comparison with previous work is fair. We showcase how the cutoff radius influences the prediction accuracy by one example in Supplementary Information Sec. 3.12. In both cases, the interaction type between any two nodes remains unchanged over time, irrespective of whether the underlying graph topology changes or not.

### Collective Relational Inference (CRI)

CRI is tailored for interacting systems where each node maintains a consistent neighborhood structure over time. The number of neighbors of $v_i$ is denoted as $|\Gamma(i)|$. The framework, illustrated in Fig. 7, can be considered as a generative model[29]. Its primary function is to predict

observed node states, specifically by predicting the state increment $\ddot{\mathbf{x}}_i^t$. This increment is used to update the nodes' states from time $t$ to $t+1$. In the context of time series in causality discovery, the state increment represents the difference between time series values. In a particle system, the state increment is analogous to acceleration. The ground-truth increment is computed based on the states of two consecutive time steps.

We assign each edge $e_{ij}$ a latent categorical random variable $z_{i,j}$. $p(z_{i,j} = z)$ is then the probability of $e_{i,j}$ having interaction type $z$ ($z = 1, 2, ..., K$). Rather than inferring $p(z_{i,j})$ for different edges independently, we consider the subgraph $S_{(i)}$ ($v_i \in V$) spanning a center node $v_i$ and its neighbors $\Gamma(i)$ as an entity. We use the random variable $z_{(i)}$ to represent the realization of the edge type of the subgraph $S_{(i)}$, which is the combination of realizations of the edge types for all edges in $S_{(i)}$. The probability $p(z_{(i)})$ captures the joint distribution of the realizations for all edges in $S_{(i)}$. We use $\phi_{z_{(i)}}(j) \in \{1, 2, ..., K\}$ to denote the interaction type $z_{i,j}$ of edge $e_{i,j}$ given the realization $z_{(i)}$ of subgraph $S_{(i)}$. For example, in Fig. 7b, $z_{(1)} = r2$ corresponds to $\{\phi_{z_{(1)}}(2) = 1, \phi_{z_{(1)}}(3) = 2)\}$, assuming that the color blue indicates type 1 and green indicates type 2. Given the edge type configuration $z_{(i)}$ of the subgraph, we adapt the standard message-passing GNN used in ref. 7 or PIG'N'PI[19] to incorporate different interaction types to predict the state increment of the center node $v_i$. Specifically, $K$ different neural networks $NN_{\theta_1}^1$, $NN_{\theta_2}^2$, ..., $NN_{\theta_K}^K$ are used to learn $K$ different interactions. Here, we consider the same architecture but different sets of parameters for these neural networks, and hence denote them as $NN^1$, $NN^2$, ..., $NN^K$. The learnable parameters in these $K$ neural networks are denoted as $\Theta = \{\theta_1, \theta_2, ..., \theta_K\}$. The predicted state increment $\hat{\ddot{\mathbf{x}}}_{i|z_{(i)}}^t$ ($\forall i$)

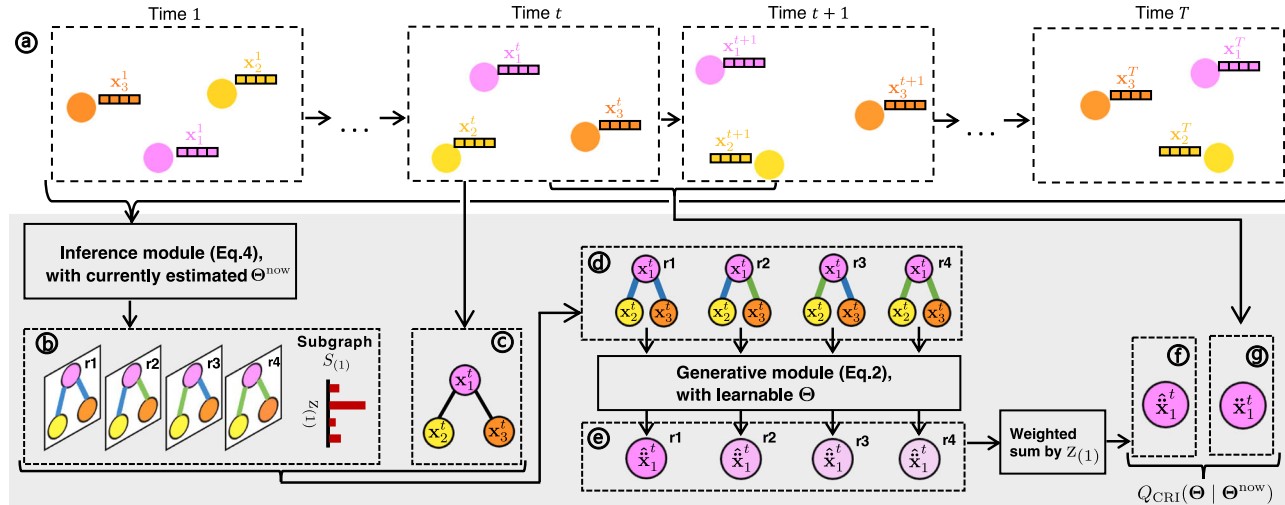

**Fig. 7 | Framework of CRI.** The proposed CRI method, shown in the gray area, takes node states at every time step and predicts the state at the next step. Dashed squares represent objects. Solid squares represent various operators, such as linear algebra operations or a neural network. To simplify, a case featuring only two distinct types of interactions is presented, though the proposed method is general and applicable to broader scenarios. **a** The interacting system over time. At every time step, each node is described by the feature vector $\mathbf{x}_i^t$ representing its state. **b** All possible realizations denoted by the random variable $z_{(1)}$ for the subgraph $S_{(1)}$.

**c** The subgraph $S_{(1)}$ at time $t$. **d** The subgraph $S_{(1)}$ exhibits different realizations at time $t$, serving as the input to the generative model. **e** The predicted state increment of $v_1$ across different realizations at time $t$. The increment is used to predict the state at the subsequent time step. **f** The final predicted state increment, representing the expectation derived from the estimated probability $z_{(1)}$. **g** The ground-truth state increment, computed from the observed states between two consecutive time steps.

given $z_{(i)}$ and the current states of nodes is computed by Eq. (1):

$$
\begin{aligned}
\text{Using standard message-passing GNN::} \quad & \hat{\mathbf{x}}_{i|z_{(i)}}^t = \mathrm{NN}_{\mathrm{node}}\left(\sum_{j\in\Gamma(i)}\mathrm{NN}^{\phi_{z_{(i)}}(j)}(\mathbf{x}_i^t,\mathbf{x}_j^t),\mathbf{x}_i^t\right) \\
\text{Or using PIG'N'PI for particles:} \quad & \hat{\mathbf{x}}_{i|z_{(i)}}^t = \sum_{j\in\Gamma(i)}\mathrm{NN}^{\phi_{z_{(i)}}(j)}(\mathbf{x}_i^t,\mathbf{x}_j^t)/m_i
\end{aligned}
\tag{1}
$$

where $\mathrm{NN}_{\mathrm{node}}$ is a neural network that takes the incoming interactions and the state of itself as input.

We use Gaussian mixture models[30] to represent the probability of the ground-truth state increment, in a manner where every realization of the subgraph is considered a component, and the latent variable $z$ determines the probability of belonging to each component. Essentially, this model is akin to Gaussian Mixtures, utilizing a neural network to shape the distribution of each component. Specifically, the conditional likelihood given the subgraph realization $z_{(i)}$ is computed by fitting the ground-truth state increment into the multivariate normal distribution whose center is the predicted state increment of the generative module, as expressed by

$$
l(\Theta|\ddot{\mathbf{x}}_i^t,z_{(i)}) = p(\ddot{\mathbf{x}}_i^t|\Theta,z_{(i)}) = \mathcal{N}\left(\ddot{\mathbf{x}}_i^t|\hat{\mathbf{x}}_{i|z_{(i)}}^t,\sigma^2\mathbf{I}\right)
\tag{2}
$$

where $\sigma^2$ is the pre-defined variance for the multivariate normal distributions.

We denote the prior probability of any subgraph having realization $z$ by $\pi_z = p(z_{(i)}=z)$ ($\forall i$). $\Upsilon$ is the set of all possible realizations of the subgraph. If all nodes have the same number of neighbors, $|\Upsilon|$ is equal to $K^{|\Gamma(i)|}$ ($\forall i$). The prior distribution $\boldsymbol{\pi} = \{\pi_1,\pi_2,...,\pi_\Upsilon\}$ and the neural network parameters $\Theta$ are the learnable parameters, which are denoted by $\boldsymbol{\Theta} = (\Theta,\boldsymbol{\pi})$.

We infer unknown parameters $\boldsymbol{\Theta}$ by maximum likelihood estimation over the marginal likelihood given the ground-truth state

increments following:

$$
L(\boldsymbol{\Theta}) = \prod_{i=1}^{|V|}\sum_{z=1}^{|\Upsilon|}\underbrace{p(z_{(i)}=z)}_{\pi_z}\prod_t l(\Theta\mid\ddot{\mathbf{x}}_i^t,z_{(i)}=z)
\tag{3}
$$

Directly optimizing $\log L(\boldsymbol{\Theta})$ in Eq. (3) with respect to $\boldsymbol{\Theta} = (\Theta,\boldsymbol{\pi})$ is intractable because of the summation in the logarithm. Therefore, we design the inference model under the generalized EM framework[15], which is an effective method to find the maximum likelihood estimate of parameters in a statistical model with unobserved latent variables. Overall, the EM iteration alternates between the expectation (E) step, which computes the expectation of the log-likelihood evaluated using the current estimation of the parameters (denoted Q function), and the maximization (M) step, which updates the parameters by maximizing the Q function found in the E step.

In the expectation (E) step, we compute the posterior probability of $z_{(i)}$ given the ground-truth state increment and the current estimation of the learnable parameters $\boldsymbol{\Theta}^{\mathrm{now}}$ by applying Bayes' theorem:

$$
\begin{aligned}
p(z_{(i)}=z|\ddot{\mathbf{x}}_i^{1:T},\boldsymbol{\Theta}^{\mathrm{now}}) &= \frac{p(z_{(i)}=z,\ddot{\mathbf{x}}_i^{1:T}|\Theta^{\mathrm{now}})}{\sum_{z'}p(z_{(i)}=z',\ddot{\mathbf{x}}_i^{1:T}|\Theta^{\mathrm{now}})} \\
&= \frac{\pi_z^{\mathrm{now}}\prod_t p(\ddot{\mathbf{x}}_i^t|\Theta^{\mathrm{now}},z_{(i)}=z)}{\sum_{z'}\pi_{z'}^{\mathrm{now}}\prod_t p(\ddot{\mathbf{x}}_i^t|\Theta^{\mathrm{now}},z_{(i)}=z')}
\end{aligned}
\tag{4}
$$

where $p(\ddot{\mathbf{x}}_i^t|\Theta^{\mathrm{now}},z_{(i)}=z)$ is computed by Eq. (2). With the posterior $p(z_{(i)}|\ddot{\mathbf{x}}_i^{1:T},\boldsymbol{\Theta}^{\mathrm{now}})$, the $Q$ function of CRI becomes:

$$
\begin{aligned}
Q_{\mathrm{CRI}}(\boldsymbol{\Theta}|\boldsymbol{\Theta}^{\mathrm{now}}) = &\sum_{i=1}^{N}\mathbb{E}_{z_{(i)}\sim p(z_{(i)}|\ddot{\mathbf{x}}_i^{1:T},\boldsymbol{\Theta}^{\mathrm{now}})}\left[\log\pi_{z_{(i)}}\right] \\
&+ \sum_{i=1}^{N}\mathbb{E}_{z_{(i)}\sim p(z_{(i)}|\ddot{\mathbf{x}}_i^{1:T},\boldsymbol{\Theta}^{\mathrm{now}})}\left[\sum_{t=1}^{T}\log l(\Theta|\ddot{\mathbf{x}}_i^t,z_{(i)})\right]
\end{aligned}
\tag{5}
$$

In the maximization (M) step, we update the prior $\boldsymbol{\pi}$ and $\Theta$ by maximizing $Q_{\mathrm{CRI}}(\boldsymbol{\Theta}|\boldsymbol{\Theta}^{\mathrm{now}})$. Note that $\boldsymbol{\pi}$ has an analytic solution but $\Theta$

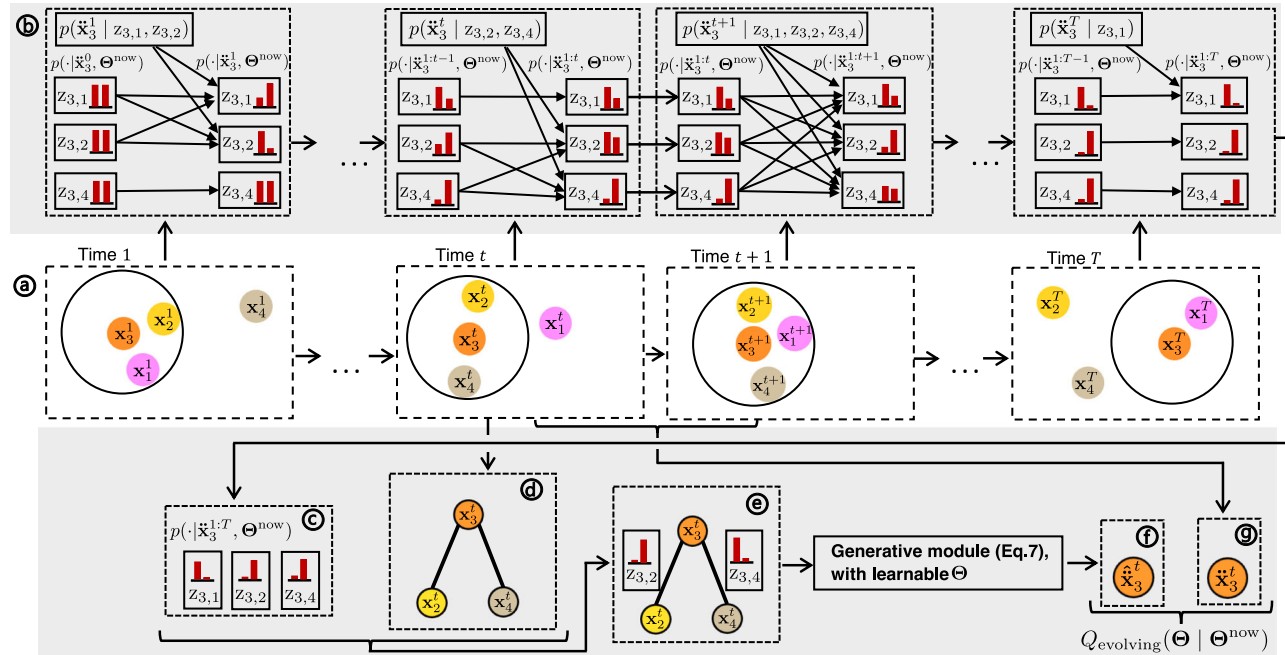

**Fig. 8 | Framework of Evolving-CRI. a** The interacting system at various moments in time. Nodes may interact with different neighbors at different times. The interaction radius of node 3 (orange) is indicated by a black circle. The feature vector $\mathbf{x}_i^t$ of each node $v_i$ contains its state at each time step $t$. **b** At each time step, we update the estimation of the posterior distributions of the interaction type for edges appearing at this time step, using Eq. (11). **c** The estimated posteriors after observing the system across all time steps. **d** Example subgraph $S_{(3)}$ at time $t$. **e** The example subgraph with different realizations of edge types at time $t$, which is the input for the generative model. **f** The predicted state increment, which is the expectation over the inferred posterior distribution in (**c**). **g** The ground-truth state increment computed from the observed states between two consecutive time steps.

does not (see Supplementary Information Sec. 2.2 for details). We take one gradient ascent step to update $\theta_1, \theta_2, ..., \theta_K$.

We iteratively update the posterior probabilities of different realizations for each subgraph in the E step and the learnable parameters $\Theta$ in the $K$ different edge neural networks and the priors $\tau$ in the M step. Convergence to the (local) optimum is guaranteed by the generalized EM procedure[15]. After training, $\mathrm{NN}^1, ..., \mathrm{NN}^K$ approximate $K$ different pairwise interaction functions. By finding the most probable realization of edge types in each subgraph, the interaction type for every edge is determined by the $\phi$ mapping. The detailed derivation and implementation of CRI are provided in Supplementary Information Sec. 2.2.

It should be noted that due to the exact computation of the expectation, the computational complexity $\mathcal{O}(N \cdot K^{|\Gamma|})$ ($|\Gamma|$ is the number of neighbors of each node) of CRI limits its application to systems with sparse interacting nodes. However, any compatible inference method can be built into CRI to approximate the expectation. To demonstrate the flexibility of CRI, we use, for instance, the basic form of variational method[30] to approximate the expectation in $Q(\Theta|\Theta^{\mathrm{now}})$. The derivation and results of this CRI variant named the Variational Collective Relational Inference (Var-CRI) are discussed in Supplementary Information Sec. 2.1. Other potential options of inference methods include advanced variational approximation methods[18,31] and Markov Chain Monte Carlo (MCMC) techniques[32].

**Evolving Collective Relational Inference (Evolving-CRI)**
The basic form of CRI presented in "Collective Relational Inference" is tailored for relational inference in which nodes consistently interact with the same neighbors. However, in various real-life scenarios, nodes may interact with varying neighbors at different times, causing the underlying graph topology to change over time. To address the challenge of inferring relations in systems with evolving graph topology, we adapt CRI and develop a new algorithm called Evolving-CRI, as shown in Fig. 8. Similarly as in CRI, we use the random variable $z_{ij} \in K$

to represent the interaction type of $e_{i,j}$. The fundamental concept behind Evolving-CRI involves updating the posterior distribution over $z_{i,j}$ of a newly appearing edge by marginalizing out the posterior distribution of all other appearing edges. As a result, the interaction type inferred for each edge captures the correlation with other incoming edges, which collectively influence the node's states. It is worth noting that our proposed approach for relational inference with evolving graph topology is different from the concept of *dynamic relational inference*[33–35] where the interaction type between two nodes can change over time. In our case, the interaction type of any edge remains the same over time, but the edges may not always exist in the underlying interaction graph.

Here, we denote the neighbors of $v_i$ at time $t$ by $\Gamma^t(i)$ and all neighbors of $v_i$ across all time steps by $\Gamma(i) = \bigcup_t \Gamma^t(i)$. Following the approach of CRI (Eq. (1)), the predicted state increment of $v_i$ ($\forall i$) at time $t$ given the edge types and the current states of those nodes within the cutoff radius is computed by Eq. (6):

Using standard message-passing GNN: $\hat{\ddot{\mathbf{x}}}_{i|z_{i,1},...,z_{i,|\Gamma^t(i)|}}^t = \mathrm{NN}_{\mathrm{node}}\left(\sum_{j \in \Gamma^t(i)} \mathrm{NN}^{z_{i,j}}(\mathbf{x}_i^t, \mathbf{x}_j^t), \mathbf{x}_i^t\right)$

Or using PIG'N'PI for particles: $\hat{\ddot{\mathbf{x}}}_{i|z_{i,1},...,z_{i,|\Gamma^t(i)|}}^t = \sum_{j \in \Gamma^t(i)} \mathrm{NN}^{z_{i,j}}(\mathbf{x}_i^t, \mathbf{x}_j^t)/m_i$

$$(6)$$

To compute the conditional likelihood given the different realizations of the edge types, we fit the ground-truth state increment by the multivariate normal distribution with the predicted state increment as the mean:

$$l(\Theta|\ddot{\mathbf{x}}_i^t, z_{i,1}, ..., z_{i,|\Gamma^t(i)|}) = p(\ddot{\mathbf{x}}_i^t|\Theta, z_{i,1}, ..., z_{i,|\Gamma^t(i)|}) = \mathcal{N}\left(\ddot{\mathbf{x}}_i^t|\hat{\ddot{\mathbf{x}}}_{i|z_{i,1},...,z_{i,|\Gamma^t(i)|}}^t, \sigma^2 \mathbf{I}\right)$$

$$(7)$$

where $\hat{\ddot{\mathbf{x}}}_{i|z_{i,1},...,z_{i,|\Gamma^t(i)|}}^t$ is computed by Eq. (6).

We denote the prior probability of any edge $e_{i,j}$ having the interaction type realization $z$ by $\tau_z = p(z_{i,j} = z)$ and the prior distribution by $\boldsymbol{\tau} = \{\tau_1, \ldots, \tau_K\}$ ($K$ is the number of different interactions). The learnable parameters in Evolving-CRI are $\boldsymbol{\Theta} = (\Theta, \boldsymbol{\tau})$.

In the expectation step, we infer by induction the posterior distribution over different interaction types of each edge given the ground-truth state increments and the current estimation of the learnable parameters $\boldsymbol{\Theta}^{\text{now}}$. At the start ($t = 0$), $p(z_{i,j}|\ddot{\mathbf{x}}_i^0, \boldsymbol{\Theta}^{\text{now}})$ is equal to the prior $\tau_{z_{i,j}}^{\text{now}}$ as there is no information available about the nodes states. Suppose that the posterior distributions $p(z_{i,j}|\ddot{\mathbf{x}}_i^{1:t-1}, \boldsymbol{\Theta}^{\text{now}})$, where $\ddot{\mathbf{x}}_i^{1:t-1}$ is the ground-truth state increment of $v_i$ until time $t-1$ for $t \geq 1$, is known, we update the posterior $p(z_{i,j}|\ddot{\mathbf{x}}_i^{1:t}, \boldsymbol{\Theta}^{\text{now}})$ for any edge $e_{i,j}$ that appears at time $t$ by the rule of sum:

$$p(z_{i,j}|\ddot{\mathbf{x}}_i^{1:t}, \boldsymbol{\Theta}^{\text{now}}) = \sum_{z_{i,-j}} p(z_{i,j}, z_{i,-j}|\ddot{\mathbf{x}}_i^{1:t}, \boldsymbol{\Theta}^{\text{now}}) \tag{8}$$

where $\sum_{z_{i,-j}}$ sums over all realizations of the other incoming edges in $S_{(i)}$ at time $t$ except for $e_{i,j}$.

The posterior distribution $p(z_{i,j}, z_{i,-j}|\ddot{\mathbf{x}}_i^{1:t}, \boldsymbol{\Theta}^{\text{now}})$ in Eq. (8) is computed by applying Bayes' theorem:

$$\begin{aligned} p(z_{i,j}, z_{i,-j}|\ddot{\mathbf{x}}_i^{1:t}, \boldsymbol{\Theta}^{\text{now}}) &\propto p(z_{i,j}, z_{i,-j}) p(\ddot{\mathbf{x}}_i^{1:t}|z_{i,j}, z_{i,-j}, \boldsymbol{\Theta}^{\text{now}}) \\ &= p(z_{i,j}, z_{i,-j}) p(\ddot{\mathbf{x}}_i^{1:t-1}|z_{i,j}, z_{i,-j}, \boldsymbol{\Theta}^{\text{now}}) p(\ddot{\mathbf{x}}_i^t|z_{i,j}, z_{i,-j}, \boldsymbol{\Theta}^{\text{now}}) \\ &\propto p(z_{i,j}, z_{i,-j}|\ddot{\mathbf{x}}_i^{1:t-1}, \boldsymbol{\Theta}^{\text{now}}) p(\ddot{\mathbf{x}}_i^t|z_{i,j}, z_{i,-j}, \boldsymbol{\Theta}^{\text{now}}) \end{aligned} \tag{9}$$

Assuming that $p(z_{i,j}, z_{i,-j}|\ddot{\mathbf{x}}_i^{1:t-1}, \boldsymbol{\Theta}^{\text{now}})$ is fully factorized, we find:

$$p(z_{i,j}, z_{i,-j}|\ddot{\mathbf{x}}_i^{1:t}, \boldsymbol{\Theta}^{\text{now}}) \propto p(z_{i,j}|\ddot{\mathbf{x}}_i^{1:t-1}, \boldsymbol{\Theta}^{\text{now}}) p(z_{i,-j}|\ddot{\mathbf{x}}_i^{1:t-1}, \boldsymbol{\Theta}^{\text{now}}) p(\ddot{\mathbf{x}}_i^t|z_{i,j}, z_{i,-j}, \boldsymbol{\Theta}^{\text{now}}) \tag{10}$$

Combining Eq. (8) and Eq. (10), we get

$$p(z_{i,j}|\ddot{\mathbf{x}}_i^{1:t}, \boldsymbol{\Theta}^{\text{now}}) \propto p(z_{i,j}|\ddot{\mathbf{x}}_i^{1:t-1}, \boldsymbol{\Theta}^{\text{now}}) \sum_{z_{i,-j}} p(z_{i,-j}|\ddot{\mathbf{x}}_i^{1:t-1}, \boldsymbol{\Theta}^{\text{now}}) p(\ddot{\mathbf{x}}_i^t|z_{i,j}, z_{i,-j}, \boldsymbol{\Theta}^{\text{now}}) \tag{11}$$

This shows that we can iteratively update the posterior $z_{i,j}$ of each edge $e_{i,j}$ by incorporating the conditional distribution of the ground-truth state increment at each time step (as illustrated in Fig. 8b). The conditional distribution $p(\ddot{\mathbf{x}}_i^t|z_{i,j}, z_{i,-j}, \boldsymbol{\Theta}^{\text{now}})$, which models the joint influence of incoming edges, is computed by Eq. (6) and Eq. (7). Finally, we denote the inferred edge type of each edge after observing the interacting system across all time steps in Eq. (11) by $p^*(z_{i,j}) = p(z_{i,j}|\ddot{\mathbf{x}}_i^{1:T}, \boldsymbol{\Theta}^{\text{now}})$.

The $Q$ function for the *Evolving-CRI* is

$$\begin{aligned} Q_{\text{evolving}}(\boldsymbol{\Theta}|\boldsymbol{\Theta}^{\text{now}}) = &\sum_{i=1}^{|V|} \sum_{j=1}^{\Gamma(i)} \mathbb{E}_{z_{i,j} \sim p^*(z_{i,j})} \left[ \log \tau_{z_{i,j}} \right] \\ &+ \sum_{i=1}^{|V|} \sum_{t=1}^{T} \mathbb{E}_{z_{i,1}, \ldots, z_{i,|\Gamma^t(i)|} \sim p^*(z_{i,1}), \ldots, p^*(z_{i,|\Gamma^t(i)|})} \\ &\left[ \log l(\boldsymbol{\Theta}|\ddot{\mathbf{x}}_i^t, z_{i,1}, \ldots, z_{i,|\Gamma^t(i)|}) \right] \end{aligned} \tag{12}$$

where $l(\boldsymbol{\Theta}|\ddot{\mathbf{x}}_i^t, z_{i,1}, \ldots, z_{i,|\Gamma^t(i)|})$ is computed by Eq. (7).

In the maximization step, we update the prior $\boldsymbol{\tau}$ and $\Theta$ by maximizing $Q_{\text{evolving}}(\boldsymbol{\Theta}|\boldsymbol{\Theta}^{\text{now}})$. Similar to CRI, $\boldsymbol{\tau}$ has the analytic solution but $\Theta$ does not. Therefore, we take one gradient ascent step to update $\theta_1, \theta_2, \ldots, \theta_K$. Finally, for verification, let us consider the case of having no observations of the interacting systems. In this case, the second term in Eq. (12) becomes 0, and $Q_{\text{evolving}}(\boldsymbol{\Theta}|\boldsymbol{\Theta}^{\text{now}})$ corresponds to the entropy because $p^*(z_{i,j})$ becomes $\tau_{z_{i,j}}$. Therefore, maximizing $Q_{\text{evolving}}$ is equivalent to maximizing the entropy, which, by the principle of maximum entropy, leads to $1/K$ probability for each edge to have any kind of interaction. This shows that in the absence of information, this method converges to a fully random estimation of the edge type, as expected.

## Performance evaluation metrics

We use the commonly used accuracy for the evaluation metric of causality discovery. For particle systems, the performance evaluation focuses on three aspects. First, the supervised learning performance is assessed through the mean absolute error $\text{MAE}_{\text{state}}$, which quantifies the discrepancy between the predicted particle states (i.e., position and velocity) and the corresponding ground-truth states. Second, we assess the ability of the relational inference methods to correctly identify different interactions. We use the permutation invariant accuracy as the metric, which is given by:

$$\text{Accuracy} = \max_{\alpha \in \Omega} \frac{1}{|E|} \sum_{e \in E} \delta(\alpha(\hat{z}(e)), z(e)) \tag{13}$$

where $\alpha$ is a permutation of the inferred interaction types and $\Omega$ is a set containing all possible permutations. The Kronecker delta $\delta(x, y)$ equals 1 if $x$ is equal to $y$ and 0 otherwise. $\hat{z}(e) \in K$ is the predicted interaction type for the edge $e$ and $z(e) \in K$ is the ground-truth interaction type of $e$. This measure accounts for the permutation of the interaction type label because good accuracy is achieved by clustering the same interactions correctly. Third, we assess the extent to which the learnt pairwise forces are consistent to the underlying physics laws. This evaluation involves two aspects: (1) how well the predicted pairwise forces approximate the ground-truth pairwise forces, which is measured by the mean absolute error on the pairwise force $\text{MAE}_{\text{ef}}$, and (2) whether the predicted pairwise forces satisfy Newton's third law, which is measured by the mean absolute value of the error in terms of force symmetry $\text{MAE}_{\text{symm}}$. To compute the predicted pairwise force required for $\text{MAE}_{\text{ef}}$ and $\text{MAE}_{\text{symm}}$, we use the generative module (i.e., the decoder) of each model that corresponds to the ground-truth interaction type, given the permutation used to compute the accuracy in Eq. (13). Therefore, $\text{MAE}_{\text{ef}}$ and $\text{MAE}_{\text{symm}}$ reflect the quality of the trained generative module, independent of the performance of the edge type prediction.

## Details of the considered case studies

The datasets used in the causality discovery experiment are from the original papers[8,11]. For the interacting particle systems, we generate the data ourselves using numerical simulations. The key distinctive property of the simulated interacting particle systems is that the inter-particle interactions are heterogeneous. Previous works, such as[7], have used some of the selected cases. However, in our study, we modified some configurations to make them more challenging and realistic.

- *Spring simulation*: Particles are randomly connected by different springs with different stiffness constants and balance lengths. Suppose $v_i$ and $v_j$ are connected by a spring with stiffness constant $k$ and balance length $L$, the pairwise force from $v_i$ to $v_j$ is $k(r_{ij} - L)\boldsymbol{n}_{ij}$ where $r_{ij} = \|\mathbf{r}_j - \mathbf{r}_i\|$ is the Euclidean distance and $\boldsymbol{n}_{ij} = \frac{\mathbf{r}_j - \mathbf{r}_i}{\|\mathbf{r}_j - \mathbf{r}_i\|}$ is the unit vector pointing from $v_i$ to $v_j$. The spring N5K2 simulation and spring N10K2 simulation have two different springs with $(k_1, L_1) = (0.5, 2.0)$ and $(k_2, L_2) = (2.0, 1.0)$. The spring N5K4 simulation has four different springs: $(k_1, L_1) = (0.5, 2.0)$, $(k_2, L_2) = (2.0, 1.0)$, $(k_3, L_3) = (2.5, 1.0)$ and $(k_4, L_4) = (2.5, 2.0)$.

- *Charge simulation*: We randomly assign electric charge $q = +1$ and $q = -1$ to different particles. The electric charge force from $v_i$ to $v_j$ is $-cq_iq_j\boldsymbol{n}_{ij}/r_{ij}^2$ where the constant $c$ is set to 1. To prevent any zeros in the denominator of the charge force equation, we add a small number $\delta$ ($\delta = 0.01$) when computing the Euclidean distance. Since particles have different charges, the system contains

attractive and repulsive interactions. Note that we do not provide charge information as an input feature for the ML algorithms. Thus, the relational inference methods need to infer whether each interaction is attractive or repulsive.

- *Crystallization simulation*: The crystallization simulation contains two different kinds of particles with local interaction, i.e. interactions only affect particles within a given proximity to each other. Hence, the underlying graph topology changes over time. In this simulation, the Lennard-Jones potential, which is given by $V_{LJ}(r) = 4\epsilon_{LJ}\{(\sigma_{LJ}/r)^{12} - (\sigma_{LJ}/r)^6\}$, exists among all nearby particles. We set $\sigma_{LJ} = 0.3$ and $\epsilon_{LJ} = 10^{-5}$. Additionally, particles of the same type have an attractive dipole-dipole force, whose potential is $V_A(r) = -Cr^{-4}$, and particles of different types have a repulsive dipole-dipole force, whose potential is $V_R(r) = Cr^{-4}$. We set the constant $C = 0.02$. To summarize, the pairwise interaction of two particles of the same and different types is governed by $V_{LJ} + V_A$ and $V_{LJ} + V_R$, respectively. The heterogeneous system contains 100 particles in total, each with the same unit mass. The simulation is adapted from[25].

Additionally, unlike the simulations in ref. 7, particles in the spring and charge simulations have varying masses. The mass $m_i$ of particle $v_i$ is sampled from the log-uniform distribution within the range $[-1, 1]$ ($\ln(m_i) \sim \mathcal{U}(-1,1)$). The initial locations and velocities of particles are both drawn from the standard Gaussian distribution $\mathcal{N}(0,1)$. We use dimensionless units for all simulations as the considered learning algorithms are not designed for any specific scale. The presented cases serve as proof of concept to evaluate the relational inference capabilities for heterogeneous interactions.

The Spring N5K2, Spring N5K4, Spring N10K2 and Charge N5K2 cases in "Relational inference with known constraints about the interacting system" each comprise 12 k simulations in total. Each simulation consists of 100 time steps with a step size 0.01. Of these 12 k simulations, 10 k are reserved for training (we train the models with 100, 500, 1 k, 5 k and 10 k simulations to assess the data efficiency), 1 k for validation and 1 k for testing. In each simulation, particles interact with all other particles and the interaction type between any particle pair remains fixed over time.

The crystallization simulation contains a single simulation of 100 particles. We generate this simulation over 500 k time steps using step size $10^{-5}$, and then downsample it to every 50 time steps, ultimately yielding a simulation with 10 k time steps. Note that it is possible to consider advanced sampling strategies (e.g., ref. 36) to sample informative time steps, but we leave this for future exploration. We use two different ways to split the simulation for training, validation and testing. First, to evaluate the interpolation ability, we randomly split the 10 k simulation steps into the training dataset, validation dataset and testing dataset with the ratio 7 : 1.5 : 1.5. Then, to evaluate the extrapolation ability, we use the first 7 k time steps as training set, next 1.5 k consecutive time steps for validation and the remaining 1.5 k time steps for testing. In the crystallization simulation, particles interact with nearby particles within a cutoff radius. However, to simplify the input for the ML methods, we constrain each particle to interact with its five closest neighbors. In the simulation, 500 edges are active at each time step, and a fixed-size tensor variable in PyTorch can represent the activated edges. It is important to note that the relational inference methods, such as CRI and the baselines, can handle varying edge sizes at different time steps. However, for the purposes of this study, the described simulation is suitable as a proof of concept and provides a straightforward implementation of the relational inference methods.

We train the relational inference models on the training dataset, fine-tune hyperparameters and select the best trained model based on the performance on the validation dataset with respect to the training objective $MAE_{state}$. It should be noted that the models are not chosen based on the metrics on which they will later be evaluated since we cannot access the ground-truth interactions during training. We then evaluate the performance of the selected trained model on the testing dataset. For the generalization evaluation in "Relational inference with known constraints about the interacting system", we train and validate the model using the training and validation datasets of Spring N5K2, and report the performance of the trained model on the testing dataset of Spring N10K2.

## Configurations of CRI, Var-CRI and Evolving-CRI

We use the same hyperparameters for CRI, Var-CRI and Evolving-CRI in each experiment. We find that the performance is mostly affected by the number of hidden layers in PIG'N'PI and the Gaussian variance $\sigma^2$. We perform grid search to tune these two parameters for different experiments. The detailed configurations are summarized in Supplementary Information Sec. 3.1. In addition, we use the Adam optimizer[37] with a learning rate 0.001 for training. All models are trained over 500 epochs.

We run experiments on a server with RTX 3090 GPUs. Each experiment is run on a single GPU−we did not implement multi-GPU parallelism. It takes nine hours for CRI to reproduce the original simulation dataset provided in the NRI reference, which is the experiment reported in Supplementary Information Sec. 3.2. Each VAR dataset in "Relational inference for causality discovery" takes three hours and the Netsim dataset takes 2.5 h (while the graph size of Netsim is larger than that of VAR, Netsim needs fewer training epochs). The time needed for the particle simulations in "Relational inference with known constraints about the interacting system" varies from one hour for five particles with 100 training examples to 47 h for ten particles with 10,000 training examples. The experiment of Evolving-CRI in "Relational inference with evolving graph topology" takes 30 h.

## Configurations of baseline models

In the causality discovery experiment, we use the same setting of NRI as suggested in the original papers for the VAR dataset[11] and for the Netsim dataset[8]. The source code of NRI for these two datasets is publicly available[8,11]. We noticed that NRI is very likely to get stuck in local minima in the VAR experiment and the random seed has a significant influence on the performance. We discard the random seeds that lead to severely inferior prediction accuracy.

For the spring and charge systems, we use the default setting of NRI and MPM as suggested in their original papers[7,12], i.e. the encoder uses a multi-layer perception to learn the initial edge embedding for the spring dataset, and a convolutional neural network (CNN) with the attention mechanism for the charge dataset. Additionally, to ensure a fair comparison, the decoder of NRI-PIG'N'PI and MPM-PIG'N'PI is the same as the one applied in CRI (including the same hidden layers and the same activation function).

For the crystallization experiment, modifications of NRI and MPM are required to learn heterogeneous systems with evolving graph topology, as discussed in "Relational inference with evolving graph topology". The CNN reducer in the original NRI and MPM first learns the initial edge embedding, and then uses additional operations to learn the edge types based on this embedding. The edge embedding is learnt by taking the states of two particles across all time steps. We modify the encoder such that only the time steps when the edge appears contribute to the edge embedding. As for the decoder, we mask the effective edges of each node at each time step and only aggregate these active edges as the incoming messages.

## Data availability

The data used in causality discovery experiments are open benchmarks provided by their original papers[8,11]. The simulation data used for learning heterogeneous inter-particle interactions have been deposited in ETH Research Collection and can be downloaded from https://www.research-collection.ethz.ch/handle/20.500.11850/610139.

## Code availability

The implementation of the proposed method is based on PyTorch[38]. The source code is available on Gitlab: https://gitlab.ethz.ch/cmbm-public/toolboxes/cri.

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

## Acknowledgements

We thank Dr. Jiawen Luo, Dr. Yuan Tian and Flavio Lorez for helpful discussions. This project has been funded by ETH Grant (no. ETH-12 21-1): Z.H., O.F. and D.S.K., and the Swiss National Science Foundation (SNSF) Grant (no. PP00P2_176878): O.F.

## Author contributions

Z.H., O.F. and D.S.K. conceptualized the idea, Z.H. developed the methodology, Z.H., O.F. and D.S.K. conceived the experiments, Z.H. conducted the experiments, Z.H., O.F. and D.S.K. analyzed the results, and all authors wrote the manuscript.

## Competing interests

The authors declare no competing interest.
