## [Peer Review File · Nature Communications]

Collective Relational Inference for learning heterogeneous interactionsREVIEWER COMMENTS

Reviewer #1 (Remarks to the Author):

Review

Title: Collective Relational Inference for learning physics-consistent heterogeneous particle interactions

Decision: Major Revision

- Summary

This paper aims at revealing particle interaction laws via a data-driven and physics-induced method called Collective Relational Inference (CRI). Instead of independently inferring the pairwise interactions, i.e., edges, CRI treats all incoming edges to a target node as a whole, learns the joint distribution of edges, and infers the edge types collectively. Based on a previous work PIG'N'PI[8] of the authors, CRI is able to ensure the physical consistency of the revealed interactions. CRI be extended to deal with dynamic interaction graphs. Experimental results on different simulation datasets show that CRI is data-efficient, generalizes better to larger systems, is more accurate when the number of interaction types increases, and is better at inferring evolving graph topology. The mean absolute errors in terms of predicted states, accelerations, and pairwise forces show that the predictions are physically consistent.

- Strengths

1. This paper is clearly written and easy to follow.
2. CRI is novel, well-formulated, technically solid, and highly flexible to integrate more computationally efficient inference methods.
3. CRI is a probabilistic framework that collectively infers the interactions within a subgraph, which may motivate subsequent research to develop more accurate relational inference methods. CRI preserves physics-consistency via delicately designed graph networks, which will draw the researchers' attention to developing methods for discovering fundamental interaction laws other than just making accurate predictions.
4. This paper reports the mean and standard deviation of all metrics in the experiments. The results are statistically significant.
5. The experiments are comprehensive. The experimental settings are described in detail.

The empirical results of interaction inference and state prediction are convincing, and the analysis can well support the conclusions.

- Weakness

1. The authors claim that CRI can learn physics-consistent interaction law for multiple interactions. Metrics like MAE_ef and MAE_symm to measure the physics-consistency of CRI. However, what interaction law is discovered is under-explored. Can the authors provide more discussions or experimental analysis? Is it possible to recover the equations of force or other quantities in the particle systems described in Section 5.5?

2. Currently, only quantitative metrics are reported to validate the effectiveness of CRI. However, it would be interesting to include some visualization to demonstrate the advantages of CRI. For example, conduct a case study to show how collective inference is more accurate than independent inference.

- Questions

1. In Line 303, the authors state that each particle only interacts with other particles within a cutoff radius. This assumption reduces the computational overhead in large-scale systems. The cutoff radius is predefined in the simulation (cf. Line 495), but how to determine the radius for CRI/Evolving-CRI during the training and testing stages?

2. Furthermore, the sensitivity of different types of forces may be different w.r.t. the distance between particles, e.g., the forces in the spring system and the charge system. When these forces coexist in a system, how to decide an appropriate cutoff radius that best tradeoff the accuracy and computational cost?

3. Line 364 shows that the computational complexity grows polynomially w.r.t. the number of edge types K . How to deal with this issue when K is relatively large? Can this problem be resolved by developing some variational methods?

4. Data efficiency is an intriguing characteristic of CRI. Which module is the key factor, the probabilistic inference module that infers the edges collectively, or the generative module that preserves physical consistency? Can the authors provide more analysis or discussions apart from the analysis in Section 3.1.1 and Section 3.1.5?

- Suggestions

1. In Line 130, the authors state that ModularMeta[12] is not selected as a baseline since it shares the same encoder with NRI[9]. However, ModularMeta is claimed to be data efficient[12] because it leverages a model-based algorithm to jointly infer the edges, which is supported by experimental results (cf. Figure 3 in [12]). Since the authors also conduct experiments to verify the data efficiency of CRI in Section 3.1.1 and Section 3.1.5, comparing CRI with ModularMeta would make the conclusion more convincing. If the comparison is not possible or unnecessary, please provide more reasons or discussions.
2. Please consider adding brackets for quantities to be taken expectation in Eq. (5), Eq. (12), and Eq. (18).
3. Suggestions for the references.
 - (1) Please check the case of the conferences and journals in the references, including [2][16][18][19][20][22][24][26][30].
 - (2) Please add pages to Reference [12].
 - (3) Please cite the officially published versions of Reference [27] and [28].

Reviewer #2 (Remarks to the Author):

- The technical contribution involves mixing the literature of neural relational inference with the authors' past work of physics-induced graph networks, and adding the possibility of variable connectivity over time (change of existence of edges, not the type of edges themselves).
- On the experimental front, this paper expands on known benchmarks for neural relational inference proposing minor variants on the evaluation. The one I particularly liked was testing the generalization from 5 Springs to 10 Springs.
- The authors also created a new interesting benchmark (Lennard-Jones interactions) for graphs with edges that appear and disappear, thus changing topology. This highlights what I believe is the main deviation from the NRI literature.
- The paper is generally well-written. I would probably make figures 1-3 a bit less verbose. Furthermore, IMO, the PIG'N'PI architecture is not that clearly explained within the main text, both as why it is required as well as how it works.
- It's worth highlighting that results in figure 4 claim a much better data efficiency than

baselines, yet Neural Relational Inference with Fast Modular Meta-learning [citation 12] claims similarly big gains on data efficiency (column A) and similar results on the charges dataset, but those results aren't included. My understanding is that both papers get the data-efficiency gains from a model-based E-M inference procedure vs model-free in the case of the baselines. [12] didn't compute all the plots of figure 4 so it would probably require rerunning, but it may be worth mentioning results where available in the text, to put things in context.

- Figure 4.E2 is a bit weird that MPM_PIGNPI gets worse before getting better, with barely any uncertainty, any thoughts on possible causes?

- I did not understand whether the model is a true 'mixture' in eq 2 line 330 or just a plain single Gaussian with fixed variance.

- Overall, I'm a bit ambivalent over acceptance, possibly leaning slightly in favor. The technical novelty on top of NRI+PIG'N'PI is mostly the time varying nature. I would build more on this aspect since it's the most novel. At the same time, the paper is well-written and the experiments are well executed.

Reviewer #3 (Remarks to the Author):

Overall, the paper is very solid from a technical standpoint and demonstrates improvement with their methodology over existing methods. The main change in methodology that drives the improvement is the analysis of the system in terms of subgraphs rather than each edge independently. I think the technical way in which this was achieved seemed reasonable and performance and comparison against existing methods is exemplary. My main concern is that the methodology, while an advance, may not be of interest to a general nature communications audience in its current form. The main concerns are that it is not yet applicable (or has not yet been shown to be applicable) to real data, and that the presentation of the manuscript will be too technical/unclear to a broad readership. In addition, while the comparison with existing methods is quite extensive, the characterization of the methods performance in terms of demonstration of results (what is

learned? Is it robust to noise?) is rather minimal. As is, I think the manuscript may be more acceptable for a computational methods type journal. Below I have expanded on these concerns.

Major Revisions:

1. Apply to real data. It is unclear that the method will be of general interest to a broad audience or if it is a solid methodological result which will mostly be of interest to others developing similar methods. The distinction, to me, lies in whether the method is applicable as is to real data or whether further development is needed. While the method clearly shows improvement over existing methods, it is unclear if that improvement is sufficient to make it useable on real systems. If the method were a demonstration of a completely new class of methods then perhaps only synthetic examples would be sufficient. As is, the ability to identify heterogeneous interactions is exciting, but it seems like the data requirements are still too high (in terms of amount of data). It is also unclear how robust the method would be to even minor amounts of measurement noise.

2. Improve accessibility of the manuscript.

- Create diagrams to illustrate cases and make sure existing figures are interpretable without reading the methods sections.

- The relationship between realizations of the subgraph (in terms of classification of the interactions) and the learning of the interactions themselves was quite confusing. I had to go back and forth between main text, methods section, and supplement to get a general idea.

- There also should be some further information about what was learned besides the summary statistics in Fig 4. What did the neural network predict as the interaction as a function of distance for example? How close was that to the true function?

3. Methodology was confusing as a stand-alone description. There was not a sufficient high-level summary of what is inferred by the method for a general audience.

- It was difficult to follow the details about how PIG'N'PI was re-implemented based on what is written, without reading the original ref. In Lines 322- 326, I was confused about if the neural network notation refers to the original PIG'N'PI work or the current work. (How are NN_{θ^i} related to NN^i ??)

- I was also confused about how the probability of being in different group types was captured based on the main text. Are all possible realizations of different grouping sampled?

- Generally, the notation needs to be either summarized in one location or better connected with diagrams. What are red bars on q plots?

Response to Reviewers' comments

We thank the reviewers for their helpful comments on our work. In response, we have made adjustments to the manuscript, conducted additional experiments, and improved the figures. Notably, we have provided further clarification on the distinctions between the proposed CRI and previous relational inference methods, to address any potential confusion about the novelty and the contributions of this work.

Additionally, we would like to highlight that, based on these comments, we have expanded the scope of our manuscript. It now not only focuses on particle systems but considers interacting systems more broadly. Specifically, we have introduced a new set of experiments on causality discovery to showcase the versatility of our method on various applications involving relational inference.

In the additional experiments, the generative module of CRI remains consistent with NRI, utilizing a standard message-passing graph neural network. The superior performance of CRI compared to NRI in accurately inferring the causal relationships between entities reaffirms the advantages of our proposed collective inference method. Through these additional evaluations, we have clarified the unique contributions of our paper, with a particular focus on highlighting the superior capabilities of the proposed collective relational inference method. This expansion not only enhances the potential impact but also broadens its applicability across various fields.

In the following, we address the reviewer's comments point by point.

Reviewer 1

Summary This paper aims at revealing particle interaction laws via a data-driven and physics-induced method called Collective Relational Inference (CRI). Instead of independently inferring the pairwise interactions, i.e., edges, CRI treats all incoming edges to a target node as a whole, learns the joint distribution of edges, and infers the edge types collectively. Based on a previous work PIG'N'PI[8] of the authors, CRI is able to ensure the physical consistency of the revealed interactions. CRI be extended to deal with dynamic interaction graphs. Experimental results on different simulation datasets show that CRI is data-efficient, generalizes better to larger systems, is more accurate when the number of interaction types increases, and is better at inferring evolving graph topology. The mean absolute errors in terms of predicted states, accelerations, and pairwise forces show that the predictions are physically consistent.

Reply: We thank the reviewer for their accurate summary of our work, and appreciate their recognition of the novelty of CRI in comparison to previous works.

Strengths 1. This paper is clearly written and easy to follow. 2. CRI is novel, well-formulated, technically solid, and highly flexible to integrate more computationally efficient inference methods. 3. CRI is a probabilistic framework that collectively infers the interactions within a subgraph, which may motivate subsequent research to develop more accurate relational inference methods. CRI preserves physics-consistency via delicately designed graph networks, which will draw the researchers' attention to developing methods for discovering fundamental interaction laws other than just making accurate predictions. 4. This paper reports the mean and standard deviation of all metrics in the experiments. The results are statistically significant. 5. The experiments

are comprehensive. The experimental settings are described in detail. The empirical results of interaction inference and state prediction are convincing, and the analysis can well support the conclusions.

Reply: We thank the reviewer for their positive comments on our work. As mentioned already above, with this revision, we already broadened the scope of CRI, and thus extend its potential use beyond particle systems.

Weakness

1. The authors claim that CRI can learn physics-consistent interaction law for multiple interactions. Metrics like MAE_ef and MAE_symm to measure the physics-consistency of CRI. However, what interaction law is discovered is under-explored. Can the authors provide more discussions or experimental analysis? Is it possible to recover the equations of force or other quantities in the particle systems described in Section 5.5?

Reply: We agree with the reviewer that the learned interaction law can be further exploited. For example, it is possible to recover the explicit formula of the interaction law by applying symbolic regression on the inferred (physics-consistent) pairwise force. The concept involves using the trained model to compute the pairwise forces, which can then be used as input for any preferred symbolic regression method. To illustrate this, we apply the off-the-shelf implementation of Deep Symbolic Optimization (DSO)¹ on the inferred pairwise forces made by the trained model on springs N5K2. The equations recovered for these two springs, namely, $2.005 \times r - 2.006$ and $0.503 \times r - 1.007$, closely align with the ground-truth equations, which are $2 \times (r - 1)$ and $0.5 \times (r - 2)$, respectively.

We believe that this is a valuable addition to our manuscript. Therefore, we have included a new section in SI (Sec. S15) to discuss the discovery of the explicit form of the learned interaction law, and discussed it in the revised main manuscript as follows:

Added/Modified in the manuscript:

(In Sec. 3.2) The superior performance of CRI in inferring the interactions, even with limited data access, forms the basis for various downstream applications, such as discovering the explicit form of the governing equations. In such cases, symbolic regression (e.g., [24]) is applied to search for the best fitting symbolic expression based on the predicted pairwise force by CRI, as demonstrated in SI Sec. S15.

Added/Modified in the manuscript:

(In Sec. 4 Conclusions) The experimental results presented emphasize CRI's potential in enhancing relational inference across diverse applications, including graph structure learning and discovery of governing physical laws in heterogeneous physical systems.

¹<https://github.com/brendenpetersen/deep-symbolic-optimization>

It is crucial to note that the equation discovery relies significantly on the quality of the applied symbolic regression techniques, which are currently advancing rapidly (e.g., see the references^{2 3 4 5 6 7}). Therefore, we want to emphasize that our contribution does not lie in the use of a specific symbolic regression algorithm. Instead, our approach offers the potential for any symbolic regression method to more effectively discover the equation based on the developed CRI method.

We clarified this point as follows:

Added/Modified in the manuscript:

(In SI Sec. S15) Further, we want to emphasize that the outcome of CRI is not limited to a specific symbolic regression algorithm but offers the potential for any symbolic regression method to more effectively discover the equation based on the developed CRI method.

2. Currently, only quantitative metrics are reported to validate the effectiveness of CRI. However, it would be interesting to include some visualization to demonstrate the advantages of CRI. For example, conduct a case study to show how collective inference is more accurate than independent inference.

Reply: We thank the reviewer for this valuable suggestion. We now added visualization of the predicted force field of CRI and NRI, which clearly demonstrates the advantage of CRI over NRI and supports our previous results shown in the main text of the manuscript.

The details of the newly added visualization are shown in SI Sec. S14 and mentioned in Sec. 3.2 in the revised manuscript.

Questions 1. In Line 303, the authors state that each particle only interacts with other particles within a cutoff radius. This assumption reduces the computational overhead in large-scale systems. The cutoff radius is predefined in the simulation (cf. Line 495), but how to determine the radius for CRI/Evolving-CRI during the training and testing stages?

Reply: Thank you for this important question. Based on previous work available in the literature, we assumed that the ground-truth cutoff radius is known. This assumption allows us to focus on developing the relational inference method and to ensure that the comparison with previous work is fair.

²Udrescu, Silviu-Marian, and Max Tegmark. AI Feynman: A physics-inspired method for symbolic regression. *Science Advances* 6, no. 16 (2020): eaay2631.

³Udrescu, Silviu-Marian, et al. AI Feynman 2.0: Pareto-optimal symbolic regression exploiting graph modularity. *Advances in Neural Information Processing Systems* 33 (2020): 4860-4871.

⁴Petersen, Brenden K., et al. Deep symbolic regression: Recovering mathematical expressions from data via risk-seeking policy gradients. In *International Conference on Learning Representations (ICLR)*, 2021.

⁵Biggio, Luca, et al. Neural symbolic regression that scales. In *International Conference on Machine Learning*, pp. 936-945. PMLR, 2021.

⁶Landajuela, Mikel, et al. A unified framework for deep symbolic regression. In *35-th Advances in Neural Information Processing Systems (NeurIPS)*, 2022.

⁷Sun, Fangzheng, et al. Symbolic physics learner: Discovering governing equations via monte carlo tree search. In *International Conference on Learning Representations (ICLR)*, 2023.

Nevertheless, we would like to note that if the ground-truth cutoff radius was unknown, we could vary the cutoff radius to construct different underlying computational graphs. We expect that the value close to the ground-truth cutoff radius should result in the smallest error, allowing for the inference of the cutoff radius. However, this topic is beyond the scope of our current manuscript, and we leave it for future work.

To clarify this assumption, we now discuss the “cutoff radius” in the revised manuscript, as follows:

Added/Modified in the manuscript:

(In Sec. 5.1) In this work, we assume the ground-truth cutoff radius is known, which allows us to focus on developing the relational inference method and to ensure that the comparison with previous work is fair.

2. Furthermore, the sensitivity of different types of forces may be different w.r.t. the distance between particles, e.g., the forces in the spring system and the charge system. When these forces coexist in a system, how to decide an appropriate cutoff radius that best tradeoff the accuracy and computational cost?

Reply: One potential way to address this challenge could be by using the largest cutoff radius of different forces to construct the computational graph. In this case, we expect the generative module to predict zeros (or close to zeros) for those types of forces having a smaller cutoff radius at large distances. It can guarantee accuracy and is more computationally affordable than simply connecting each particle to all other particles. However, it is an open research question whether a more appropriate cutoff radius exists (e.g., a value between the smallest and largest cutoff radius of different forces) that trades off the accuracy and computational cost better.

3. Line 364 shows that the computational complexity grows polynomially w.r.t. the number of edge types K . How to deal with this issue when K is relatively large? Can this problem be resolved by developing some variational methods?

Reply: Yes, absolutely, this is exactly the reason why we also developed and tested the Var-CRI variant. We demonstrate via Var-CRI that it is possible to use variational approximation methods for the inference in CRI. Except for the basic variational method discussed in the manuscript, we can also consider more advanced variational methods (e.g., hierarchical variational methods ⁸) or the Markov chain Monte Carlo (MCMC) methods ⁹. However, as approximation inference techniques are not the central piece of our contribution, we limited the description and discussion of Var-CRI to the appendix.

In our revised manuscript, the methodology details of Var-CRI are explained in SI Sec. S2, and we discuss this aspect in Sec. 5.2, as follows:

⁸Ranganath, Rajesh, Dustin Tran, and David Blei. Hierarchical variational models. In International conference on machine learning, pp. 324-333. PMLR, 2016.

⁹Robert, Christian P., George Casella, and George Casella. Monte Carlo statistical methods. Vol. 2. New York: Springer, 1999.

Added/Modified in the manuscript:

(In Sec. 5.2) It should be noted that due to the exact computation of the expectation, the computational complexity $\mathcal{O}(N \cdot K^{|\Gamma|})$ ($|\Gamma|$ is the number of neighbors of each node) of CRI limits its application to systems with sparse interacting nodes. However, any compatible inference method can be built into CRI to approximate the expectation. To demonstrate the flexibility of CRI, we use, for instance, the basic form of variational method [30] to approximate the expectation in $Q(\Theta | \Theta^{now})$. The derivation and results of this CRI variant named the Variational Collective Relational Inference (Var-CRI) are discussed in SI Sec. S2. Other potential options of inference methods include advanced variational approximation methods [19, 31] and Markov Chain Monte Carlo (MCMC) techniques [32].

4. Data efficiency is an intriguing characteristic of CRI. Which module is the key factor, the probabilistic inference module that infers the edges collectively, or the generative module that preserves physical consistency? Can the authors provide more analysis or discussions apart from the analysis in Section 3.1.1 and Section 3.1.5?

Reply: This is an excellent point. In the original version of our manuscript, we compared CRI (using PIGNPI as generative module) with NRI-PIGNPI, which uses the same generative module but different inference module, and showed that the performance of NRI-PIGNPI is far behind CRI. This clearly demonstrates that the probabilistic inference module, which infers the edges collectively, is the key factor.

To further extend this comparison, we now also added a case study in which we compare CRI-GNN (which uses the collective inference module whose generative module uses the same graph neural network as NRI) and NRI-PIGNPI to CRI-PIGNPI. This means that CRI-GNN and CRI-PIGNPI differ only in the generative module, while the difference between NRI-PIGNPI and CRI-PIGNPI lies solely in the inference module. The results show that the relation prediction accuracy of CRI-GNN is almost the same as CRI-PIGNPI, which further underscores our finding that the probabilistic inference module is the determining factor. We believe that this also indicates that applying CRI on other tasks is very promising as we can change the generative module to fit better the specific task of interest.

The additional case study is provided in SI Sec. S12, with the new figure Fig. S2 as follows:

The added figure to compare NRI-PIGNPI, NRI-GNN and CRI-PIGNPI

Suggestions

1. In Line 130, the authors state that ModularMeta[12] is not selected as a baseline since it shares the same encoder with NRI[9]. However, ModularMeta is claimed to be data efficient[12] because it leverages a model-based algorithm to jointly infer the edges, which is supported by experimental results (cf. Figure 3 in [12]). Since the authors also conduct experiments to verify the data efficiency of CRI in Section 3.1.1 and Section 3.1.5, comparing CRI with ModularMeta would make the conclusion more convincing. If the comparison is not possible or unnecessary, please provide more reasons or discussions.

Reply: As described in the original ModularMeta (reference [14] in the revised manuscript) paper, it uses the same encoder as NRI to generate a proposal for the simulated annealing (SA) step (see the *Meta-learning a proposal function* part in Sec.4 of [14]). Thus, ModularMeta is akin to NRI in inferring the probability distribution of interaction types. The distinction arises in the sampling strategy after establishing the probability distribution of interaction types (see the third paragraph in Sec.4 of [14]). Since the inference part to compute the probability distribution of interaction types is of most importance to the relational inference task, and we have compared to NRI, we believe that a comparison to ModularMeta would not add substantial value to the manuscript.

We discuss this aspect in the revised manuscript as follows:

Added/Modified in the manuscript:

(In revised Sec. 1 Introduction) Alet et al. [14] used the same encoder as NRI [7] to infer the pairwise interaction types.

Added/Modified in the manuscript:

(In revised Sec. 3.2) It is worth noting that ModularMeta [14] uses the same inference module as NRI for inferring edge type, resulting in independent inference of the interaction types for different edges, and since the inference module is the key component for relational inference, we exclude ModularMeta for comparison.

Furthermore, we should mention that running experiments with ModularMeta was quite challenging due to many unexplained hard-coded values in the released code¹⁰. While it is possible to run ModularMeta on other problems, we are concerned that when applied to new datasets, such as ours, we are not able to guarantee correctness of the results, which is, in our opinion, a prerequisite for use in a published study.

2. Please consider adding brackets for quantities to be taken expectation in Eq. (5), Eq. (12), and Eq. (18).

Reply: Thanks for pointing us towards this typo. We added the brackets in expectations.

¹⁰<https://github.com/FerranAlet/modular-metalearning/tree/master/neurips2019>

3. Suggestions for the references. (1) Please check the case of the conferences and journals in the references, including [2][16][18][19][20][22][24][26][30]. (2) Please add pages to Reference [12]. (3) Please cite the officially published versions of Reference [27] and [28].

Reply: We thank the reviewer for pointing out the problems. We corrected the references accordingly.

Reviewer 2

- The technical contribution involves mixing the literature of neural relational inference with the authors' past work of physics-induced graph networks, and adding the possibility of variable connectivity over time (change of existence of edges, not the type of edges themselves).

Reply: We appreciate the reviewer's feedback, and we would like to address the concern raised regarding the technical contribution. Respectfully, we disagree with the statement that our work merely blends the neural relational inference literature with our prior research on physics-induced graph networks. It is important to highlight that even without incorporating physics-induced graph networks, the proposed CRI fundamentally diverges from Neural Relational Inference (NRI) (reference [7] in the revised paper). Specifically, CRI's approach involves treating all incoming edges of a target node *jointly*, learning the *joint distribution* of edges, and *collectively* inferring edge types. In contrast, NRI independently infers the pairwise interactions. This distinctive approach, where CRI infers interactions collectively, sets it apart from NRI and underlines its fundamental novelty.

To address this confusion about the novelty, we have adapted the structure and the content of the manuscript. It now not only focuses on particle systems but also considers interacting systems more broadly. Specifically, we have introduced a new set of experiments on causality discovery. The difference between CRI and the previously proposed NRI is further supported by the new experiment on causality discovery. The highlighted revisions can be found in the revised Abstract, Sec. 2.1, Sec. 2.2 and Sec. 3.1.

- On the experimental front, this paper expands on known benchmarks for neural relational inference proposing minor variants on the evaluation. The one I particularly liked was testing the generalization from 5 Springs to 10 Springs.

Reply: We appreciate the reviewer's acknowledgment of the generalization experiment. As mentioned above, we now also added additional evaluations on the task of causal discovery.

- The authors also created a new interesting benchmark (Lennard-Jones interactions) for graphs with edges that appear and disappear, thus changing topology. This highlights what I believe is the main deviation from the NRI literature.

Reply: We appreciate the reviewer's recognition of the new benchmark. While we agree that the proposed methodology for relational inference with an evolving graph topology extends the proposed CRI to a new level of complexity, we would like to emphasize a critical methodological aspect: the

proposed CRI differs fundamentally from NRI in that CRI collectively infers the interaction types of incoming edges, as opposed to NRI, which infers pairwise interactions independently. As we show in the manuscript, this leads to significant improvements in the accuracy and performance of our approach compared to baselines.

In the revised manuscript, we emphasize the difference between CRI and NRI multiple times, such as

Added/Modified in the manuscript:

(In Sec. 2) CRI differs from previous probabilistic methods [7, 12, 16] in how the probability distribution of the unknown interaction types is computed, as illustrated in Fig. 1.

Added/Modified in the manuscript:

(In Sec. 2.1) The inference module is designed for *collective inference* of the interaction types. This characteristic distinguishes CRI from previous relational inference methods [7, 12, 16]...

- The paper is generally well-written. I would probably make figures 1-3 a bit less verbose. Furthermore, IMO, the PIG'N'PI architecture is not that clearly explained within the main text, both as why it is required as well as how it works.

Reply: We appreciate the reviewer's recognition of the writing quality of the paper.

For figures 1-3, we agree that the illustrations of CRI and Evolving-CRI included an abundance of technical details, making them challenging to understand without reading the Method section. As a resolution, we relocated those two figures to the Method section, and introduced a new simplified figure in the main text to explain the idea of CRI at a more accessible level. We believe the current version of the figure, which is shown below, provides a clearer and more intuitive presentation.

Regarding PIG'N'PI, we would like to emphasize that it is only one potential choice for the generative module in CRI. Our main contribution in this paper lies in the relational inference module rather than the generative module. Consequently, we reference our earlier paper on PIG'N'PI without providing a detailed explanation of its architecture in the main text.

We discuss this aspect in the revised manuscript as follows:

Added/Modified in the manuscript:

(In revised Sec. 2.2) The generative module of CRI can be any type of a graph neural network, including the basic message-passing graph neural networks (see Sec. 3.1). Nevertheless, if there are known constraints about the interacting systems, they can also be integrated. For example, the generative module applied for interacting particle systems may ensure that the learnt interactions are *physics-consistent*, meaning they adhere to Newton's laws of motion. In those cases, as described in Sec. 3.2, we use the recently proposed physics-induced graph network for particle interaction (PIG'N'PI) [18].

- It's worth highlighting that results in figure 4 claim a much better data efficiency than baselines, yet Neural Relational Inference with Fast Modular Meta-learning [citation 12] claims similarly big gains on data efficiency (column A) and similar results on the charges dataset, but those results aren't included. My understanding is that both papers get the data-efficiency gains from a model-based E-M inference procedure vs model-free in the case of the baselines. [12] didn't compute all the plots of figure 4 so it would probably require rerunning, but it may be worth mentioning results where available in the text, to put things in context.

Reply: The main reason for not including ModularMeta is its reliance on the same encoder as NRI, as indicated in the original paper ([14] in the revised manuscript). ModularMeta uses the same encoder as NRI to generate a proposal for the simulated annealing (SA) step, thus aligning its methodology closely with NRI (see the *Meta-learning a proposal function* part in Sec.4 of [14]). As such, ModularMeta and NRI share the process of inferring the probability distribution of interaction types. The distinction between ModularMeta and NRI lies solely in the sampling strategy after establishing the probability distribution of interaction types, as detailed in the third paragraph in Sec.4 of [14]. As the inference part to compute the probability distribution of interaction types is of most importance to the relational inference task, and we have compared to NRI, we think it is unnecessary to compare to ModularMeta again.

As already discussed in the response to suggestion 1 of reviewer 1, we discuss this aspect in the revised manuscript as follows:

Added/Modified in the manuscript:

(In revised Introduction) Alet et al. [14] used the same encoder as NRI [7] to infer the pairwise interaction types.

Added/Modified in the manuscript:

(In revised Sec. 3.2) It is worth noting that ModularMeta [14] uses the same inference module as NRI for inferring edge type, resulting in independent inference of the interaction types for different edges, and since the inference module is the key component for relational inference, we exclude ModularMeta for comparison.

Furthermore, we should mention that running experiments with ModularMeta is quite challenging due to many unexplained hard-coded values in the released code¹¹. While it is possible to run ModularMeta on other problems, we are concerned that when applied to new datasets, such as ours, we are not able to guarantee correctness of the results, which is, in our opinion, a prerequisite for use in a published study.

- Figure 4.E2 is a bit weird that MPM_PIGNPI gets worse before getting better, with barely any uncertainty, any thoughts on possible causes?

Reply: Thank you for this interesting question. In fact, the interaction type prediction accuracy of MPM_PIGNPI and NRI_PIGNPI is almost 50% when the training data is fewer than 1000 in the Charge N5K2 case (Fig.4 E1), which is equivalent to a random guess. This implies that their generative module fails to learn any information about the actual force as the generative module depends on the predicted interaction type. Consequently, calculating the MAE_ef for MPM_PIGNPI and NRI_PIGNPI is not meaningful in this context, and their MAE_ef within this range of training input size is of no significance to us. We indicate this point in the revised manuscript, in Fig. 4 E2 and Sec. 3.2, as follows:

Added/Modified in the manuscript:

(In Sec. 3.2) We note that NRI-PIG'N'PI and MPM-PIG'N'PI only achieve about 50% accuracy in the Charge N5K2 dataset when the training data is less than 1000, which is similar to random guessing. This implies that their generative module fails to learn any information about the actual force in this range since the generative module depends on the predicted interaction type. Analyzing their learnt forces makes no sense with such a limited amount of training data, as indicated by the empty symbols and dashed lines in Fig. 4-E2.

The updated Fig. 4 E2

- I did not understand whether the model is a true 'mixture' in eq 2 line 330 or just a plain single Gaussian with fixed variance.

Reply: The model represents the "mixture model" (See Sec.2.3.9 in Christopher Bishop's Pattern Recognition and Machine Learning¹²) in a manner where different realizations of the edge type in

¹¹ <https://github.com/FerranAlet/modular-metalearning/tree/master/neurips2019>

¹² Bishop, Christopher M. Pattern Recognition and Machine Learning. New York: Springer, 2006.

each subgraph are considered as different components, and a latent variable (in our case, denoted as z) determines the probability of belonging to each component. Essentially, this model is akin to Gaussian Mixtures (See Sec.9 of Christopher Bishop's Pattern Recognition and Machine Learning), utilizing a neural network to shape the distribution of each component. Eq 2 is the probability of an observation belonging to the $z_{(i)}$ -th realization.

We explain this with more details in the revised manuscript as follows:

Added/Modified in the manuscript:

(In revised Sec.5.2) We use Gaussian mixture models [30] to represent the probability of the ground-truth state increment, in a manner where every realization of the subgraph is considered a component, and the latent variable z determines the probability of belonging to each component. Essentially, this model is akin to Gaussian Mixtures, utilizing a neural network to shape the distribution of each component. Specifically, the conditional likelihood given the subgraph realization $z_{(i)}$ is computed by fitting the ground-truth state increment into the multivariate normal distribution whose center is the predicted state increment of the generative module, as expressed by [Eq. 2].

- Overall, I'm a bit ambivalent over acceptance, possibly leaning slightly in favor. The technical novelty on top of NRI+PIG'N'PI is mostly the time varying nature. I would build more on this aspect since it's the most novel. At the same time, the paper is well-written and the experiments are well executed.

Reply: We appreciate the reviewer's positive response to our work. Nevertheless, we wish to reiterate that **our method is distinct from NRI+PIG'N'PI**. The inference technique we propose fundamentally differs from NRI as we **collectively** infer the incoming interactions of a target node. This distinction led us to name our method CRI (Collective relational inference). We have made significant changes to the Introduction and the Method sections in the revised version in an effort to eliminate any such confusion. Furthermore, in the revised manuscript, we included additional experiments on causality discovery to compare the proposed CRI (without PIG'N'PI) with NRI. In these additional experiments, the generative model of CRI is the same as NRI, utilizing a standard message-passing graph neural network. Once again, CRI outperforms NRI, demonstrating the distinction between the two models.

Reviewer 3

Overall, the paper is very solid from a technical standpoint and demonstrates improvement with their methodology over existing methods. The main change in methodology that drives the improvement is the analysis of the system in terms of subgraphs rather than each edge independently. I think the technical way in which this was achieved seemed reasonable and performance and comparison against existing methods is exemplary. My main concern is that the methodology, while an advance, may not be of interest to a general nature communications audience in its current form.

The main concerns are that it is not yet applicable (or has not yet been shown to be applicable) to real data, and that the presentation of the manuscript will be too technical/unclear to a broad readership. In addition, while the comparison with existing methods is quite extensive, the characterization of the methods performance in terms of demonstration of results (what is learned? Is it robust to noise?) is rather minimal. As is, I think the manuscript may be more acceptable for a computational methods type journal. Below I have expanded on these concerns.

Reply: We appreciate the reviewer’s support of our technical contribution, and we are convinced that the raised concerns were addressed in the revised manuscript, which is considerably more general than the initial submission, demonstrating the wide range of possible applications. We also included additional evaluations to demonstrate the robustness of the proposed methodology to noise. Additionally, we illustrated how the underlying physical equations can be deduced from the inferred interactions with symbolic regression. Each of the raised concerns are addressed below point-by-point.

1. Apply to real data. It is unclear that the method will be of general interest to a broad audience or if it is a solid methodological result which will mostly be of interest to others developing similar methods. The distinction, to me, lies in whether the method is applicable as is to real data or whether further development is needed. While the method clearly shows improvement over existing methods, it is unclear if that improvement is sufficient to make it useable on real systems. If the method were a demonstration of a completely new class of methods then perhaps only synthetic examples would be sufficient. As is, the ability to identify heterogeneous interactions is exciting, but it seems like the data requirements are still too high (in terms of amount of data). It is also unclear how robust the method would be to even minor amounts of measurement noise.

Reply: We greatly appreciate the reviewer’s comments. Our work distinctly underscores the critical importance of accounting correlations among various interactions in relational inference. The necessity for relational inference in interacting systems spans across numerous scenarios, and our emphasis on this, coupled with the proposed methodology, addresses a broad spectrum of interest. We will further elaborate on this argument in the following section.

First, the reason for utilizing existing benchmark datasets rather than real data is that we were not able to find an open-source real dataset *with the ground-truth* that would allow us to evaluate the quality of learnt interactions and compare to previous methods.

More importantly, the task of relational inference holds a wide-ranging applicability across numerous domains, including

- predicting high-energy particle decay structures in high energy physics ¹³
- understanding protein allostery ¹⁴
- representing arbitrary two-qubit systems compactly ¹⁵

¹³Kahn, James, et al. Learning tree structures from leaves for particle decay reconstruction. *Machine Learning: Science and Technology* 3.3 (2022): 035012.

¹⁴Zhu, Jingxuan, et al. Neural relational inference to learn long-range allosteric interactions in proteins from molecular dynamics simulations. *Nature communications* 13, no. 1 (2022): 1661.

¹⁵Nautrup, Hendrik Poulsen, et al. Operationally meaningful representations of physical systems in neural networks. *Machine Learning: Science and Technology* 3, no. 4 (2022): 045025.

- epilepsy seizure identification ¹⁶
- improving the performance of multi-agent reinforcement learning algorithms in both cooperative and competitive scenarios ¹⁷
- causal discovery ¹⁸

We acknowledge that the broader applicability of the proposed methodology was not sufficiently emphasized in the previous version of the manuscript. We have now restructured the manuscript to highlight the broader scope of relational inference, moving beyond the exclusive focus on interacting particle systems. In particular, we have introduced a causality discovery experiment to showcase the versatility of the proposed method. In the additional experiments, which include a semi-real fMRI dataset ¹⁹, we compare the proposed CRI (without PIG'N'PI) to NRI on predicting the causal relationship between entities. The superior performance of CRI over NRI demonstrates the advantage of CRI for the general relational task, in addition to particle systems.

Given the crucial applications of relational inference, we believe that making a significant contribution to the fundamental task of relational inference will also be of interest to a wider readership.

Finally, we would like to thank the reviewer for suggesting the experiment on measurement noise, as it signifies the potential applicability of the method in real-world scenarios. Therefore, we included an experiment to evaluate the robustness of the proposed CRI by introducing measurement noise to the Spring N5K2 dataset. The results demonstrate that CRI is robust to a certain level of noise, and its performance even becomes slightly better with small noise. Of course, it fails with severe noise. The details of this new experiment are discussed in SI Sec. S13 in the revised manuscript. The findings are highlighted here, as follows:

Added/Modified in the manuscript:

(In revised SI Sec. S13) We define the noise level as the average relative change of the target (state increment). . . The performances of CRI with different noise levels are summarized in Table S11. We find that CRI is robust to a certain level of noise. Interestingly, the performance becomes slightly better with small noise (e.g., $\beta = 1e-7$). However, it fails to infer the heterogeneous interactions with severe noise (e.g., $\beta = 1e-3$).

¹⁶Zhao, Yanna, et al. Automatic seizure identification from EEG signals based on brain connectivity learning. *International journal of neural systems* 32, no. 11 (2022): 2250050.

¹⁷Zhang, Xianjie, et al. Structural relational inference actor-critic for multi-agent reinforcement learning. *Neurocomputing* 459 (2021): 383-394.

¹⁸Löwe, Sindy, et al. Amortized causal discovery: Learning to infer causal graphs from time-series data. In *Conference on Causal Learning and Reasoning*, pp. 509-525. PMLR, 2022.

¹⁹Smith, Stephen M., et al. Network modelling methods for FMRI. *Neuroimage* 54.2 (2011): 875-891.

Performance of CRI with different noise levels (Table S11 in the revised manuscript)

β	noise level	Accuracy	MAE _{ef}	MAE _{symm}	MAE _{state} 1 step	MAE _{state} 10 step
0.0	0.0	0.9920 ± 0.0004	0.1071 ± 0.0024	0.0949 ± 0.0105	0.0029 ± 0.0001	0.0285 ± 0.0008
1e-7	0.0250	0.9930 ± 0.0017	0.1017 ± 0.0064	0.0998 ± 0.0126	0.0028 ± 0.0002	0.0274 ± 0.0019
1e-6	0.3816	0.9924 ± 0.0021	0.1051 ± 0.0069	0.0934 ± 0.0027	0.0030 ± 0.0002	0.0284 ± 0.0023
1e-5	4.3715	0.9929 ± 0.0016	0.1009 ± 0.0074	0.0967 ± 0.0092	0.0051 ± 0.0001	0.0272 ± 0.0019
1e-4	36.9642	0.9900 ± 0.0021	0.1125 ± 0.0105	0.1003 ± 0.0119	0.0393 ± 0.0000	0.0477 ± 0.0015
1e-3	277.8361	0.5128 ± 0.0159	23.1797 ± 3.4117	40.7052 ± 9.9086	0.3799 ± 0.0012	0.6010 ± 0.0355

2. Improve accessibility of the manuscript.

- Create diagrams to illustrate cases and make sure existing figures are interpretable without reading the methods sections.

Reply: We thank the reviewer for this suggestion. As already discussed in the response to reviewer 2, in the revised version, we introduced a new simplified figure (current Fig. 2 in the revised manuscript, see below) to illustrate the working pipeline of CRI. We relocated the previous figures illustrating the architectures of CRI and Evolving-CRI to their respective subsections within the Method section. This change allows readers to grasp the methodology without needing to refer to technical details beforehand.

The added simplified figure to explain the idea of CRI. (Fig.2 in the revised manuscript)

- The relationship between realizations of the subgraph (in terms of classification of the interactions) and the learning of the interactions themselves was quite confusing. I had to go back and forth between main text, methods section, and supplement to get a general idea.

Reply: The realizations of the subgraph represent all potential different configurations of interaction types among the edges in that subgraph. Learning these interactions entails approximating various interaction functions, such as the distinct force functions in a particle system using neural networks. We acknowledge that fully comprehending the presented method, particularly regarding the employed subgraph concept, requires careful consideration. In the revised manuscript, we made these concepts clear in the way that we explain the high-level idea in the Introduction and specify their exact mathematical meaning in the Method section, as follows

Added/Modified in the manuscript:

(In Sec. 2) our approach takes into account subgraphs comprising a center node and its neighboring nodes as a collective entity and infers the interaction types of the edges within each subgraph jointly.

Added/Modified in the manuscript:

(In Sec. 2) a generative module that is a graph neural network [17] approximating pairwise interactions. . .

Added/Modified in the manuscript:

(In Sec. 5.2) the realization of the edge type of the subgraph $S_{(i)}$, which is the combination of realizations of the edge types for all edges in $S_{(i)}$. The probability $p(z_{(i)})$ captures the joint distribution of the realizations for all edges in $S_{(i)}$

Added/Modified in the manuscript:

(In Sec. 5.2) K different neural networks $NN_{\theta_1}^1, NN_{\theta_2}^2, \dots, NN_{\theta_K}^K$ are used to learn K different interactions

- There also should be some further information about what was learned besides the summary statistics in Fig 4.

Reply: We thank the reviewer for this suggestion. We have included visualizations of the learnt force fields by CRI and NRI in SI Sec. S14 in the revised manuscript (See the blue figures in Fig. S3 and Fig. S4). Additionally, we visualize the disparity between the predicted force fields of different methods and the ground-truth force field (again in Sec. S14 of the revised manuscript; see the green figures in Fig. S3 and Fig. S4). These visualizations clearly show that CRI can learn the force fields better than NRI, and support our previous results shown in the main text of the manuscript.

We modified the discussion in Results to point to these additional visualizations in Sec. 3.2 of the revised manuscript, as follows:

Added/Modified in the manuscript:

(In Sec. 3.2) We find that CRI achieves a lower error in learning the pairwise force functions in all systems (see Fig. 4 middle and SI Sec. S14). . .

- What did the neural network predict as the interaction as a function of distance for example? How close was that to the true function?

Reply: We thank the reviewer for this question, which was also asked by Reviewer 1.

Yes, we can further derive the force equation or the equations of other relevant quantities by applying symbolic regression on the inferred pairwise forces. The approach involves utilizing the trained model to predict the values of pairwise forces, which then serve as input to an *arbitrary* symbolic regression method. For instance, we applied an off-the-shelf implementation of Deep Symbolic Optimization (DSO)²⁰ on the predictions made by the trained model for the springs in dataset N5K2. The derived equations for these two springs are: $2.005 \times r - 2.006$ and $0.503 \times r - 1.007$. Notably, these derived equations closely resemble the ground-truth equations, which are $2 \times (r - 1)$ and $0.5 \times (r - 2)$.

As already discussed in the response to reviewer 1, we believe that this is a valuable addition to our manuscript. Therefore, we have included a new section in SI (Sec. S15) to discuss the discovery of the explicit form of the learned interaction law, and discussed it in the revised main manuscript as follows:

Added/Modified in the manuscript:

(In Sec. 3.2) The superior performance of CRI in inferring the interactions, even with limited data access, forms the basis for various downstream applications, such as discovering the explicit form of the governing equations. In such cases, symbolic regression (e.g., [24]) is applied to search for the best fitting symbolic expression based on the predicted pairwise force by CRI, as demonstrated in SI Sec. S15.

Added/Modified in the manuscript:

(In Sec. 4 Conclusions) The experimental results presented emphasize CRI's potential in enhancing relational inference across diverse applications, including graph structure learning and discovery of governing physical laws in heterogeneous physical systems.

Moreover, it is crucial to note that the equation discovery relies significantly on the quality of the applied symbolic regression techniques, which are currently advancing rapidly (e.g., see the

²⁰<https://github.com/brendenpetersen/deep-symbolic-optimization>

references^{21 22 23 24 25 26}). Therefore, we want to emphasize that our contribution does not lie in the use of a specific symbolic regression algorithm. Instead, our approach offers the potential for any symbolic regression method to more effectively discover the equation based on the developed CRI method. We clarified this point as follows:

Added/Modified in the manuscript:

(In SI Sec. S15) Further, we want to emphasize that our contribution does not lie in the use of a specific symbolic regression algorithm. Instead, our approach offers the potential for any symbolic regression method to more effectively discover the equation based on the developed CRI method.

3. Methodology was confusing as a stand-alone description. There was not a sufficient high-level summary of what is inferred by the method for a general audience.

- It was difficult to follow the details about how PIG’N’PI was re-implemented based on what is written, without reading the original ref. In Lines 322- 326, I was confused about if the neural network notation refers to the original PIG’N’PI work or the current work. (How are NN_{theta}^l related to $NN^{i??}$)

Reply: As mentioned in previous replies, CRI is not solely tailored for particle systems; its generative module can encompass any type of graph neural networks. To clarify this further, we make a distinct separation between CRI and CRI-PIG’N’PI, where CRI refers to the probabilistic inference while CRI-PIG’N’PI is the probabilistic inference plus PIG’N’PI as generative module. We clarified this distinction in the revised manuscript, especially in Sec. 2.2, Sec. 3 and Sec. 5. PIG’N’PI was specifically chosen for particle systems to ensure adherence to physics principles. Since the primary contribution of this work lies in the inference module rather than the generative module, we solely make a reference to the original publication of PIG’N’PI ²⁷.

²¹Udrescu, Silviu-Marian, and Max Tegmark. AI Feynman: A physics-inspired method for symbolic regression. *Science Advances* 6, no. 16 (2020): eaay2631.

²²Udrescu, Silviu-Marian, et al. AI Feynman 2.0: Pareto-optimal symbolic regression exploiting graph modularity. *Advances in Neural Information Processing Systems* 33 (2020): 4860-4871.

²³Petersen, Brenden K., et al. Deep symbolic regression: Recovering mathematical expressions from data via risk-seeking policy gradients. In *International Conference on Learning Representations (ICLR)*, 2021.

²⁴Biggio, Luca, et al. Neural symbolic regression that scales. In *International Conference on Machine Learning*, pp. 936-945. PMLR, 2021.

²⁵Landajuena, Mikel, et al. A unified framework for deep symbolic regression. In *35-th Advances in Neural Information Processing Systems (NeurIPS)*, 2022.

²⁶Sun, Fangzheng, et al. Symbolic physics learner: Discovering governing equations via monte carlo tree search. In *International Conference on Learning Representations (ICLR)*, 2023.

²⁷Han, Zhichao, David S. Kammer, and Olga Fink. Learning physics-consistent particle interactions. *PNAS nexus* 1.5 (2022): pgac264.

- I was also confused about how the probability of being in different group types was captured based on the main text. Are all possible realizations of different grouping sampled?

Reply: The posterior distribution of each subgraph with different realizations is calculated using Eq.4 for CRI and Eq.11 for Evolving-CRI. Essentially, each subgraph realization represents a potential configuration of the interaction type for all edges within that specific subgraph. And, yes, all possible realizations are sampled.

In the Method part of the revised manuscript, we provided the more detailed explanation about computing the probability of different realizations, as follows:

Added/Modified in the manuscript:

(In Sec. 5.2; for CRI) We use the random variable $z_{(i)}$ to represent the realization of the edge type of the subgraph $S_{(i)}$, which is the combination of realizations of the edge types for all edges in $S_{(i)}$. The probability $p(z_{(i)})$ captures the joint distribution of the realizations for all edges in $S_{(i)}$.

Added/Modified in the manuscript:

(In Sec. 5.2; for CRI) In the expectation (E) step, we compute the posterior probability of $z_{(i)}$ given the ground-truth state increment and the current estimation of the learnable parameters Θ^{now} by applying Bayes' theorem [Eq. 4].

Added/Modified in the manuscript:

(In Sec. 5.3; for Evolving-CRI) we use the random variable $z_{i,j} \in K$ to represent the interaction type of $e_{i,j}$

Added/Modified in the manuscript:

(In Sec. 5.3; for Evolving-CRI) Combining Eq. 8 and Eq. 10, we get [Eq. 11]. This shows that we can iteratively update the posterior $z_{i,j}$ of each edge $e_{i,j}$.

- Generally, the notation needs to be either summarized in one location or better connected with diagrams. What are red bars on q plots?

Reply: We thank the reviewer for this suggestion.

In the revised manuscript, we summarize all the notations utilized in Table S1, in SI Sec. S1. We clarify this in Introduction and Method both, as follows:

Added/Modified in the manuscript:

(In the caption of Fig. 1) \mathcal{F}_v and \mathcal{F}_e represent the function approximation of the node state update function and interaction function (by neural networks), respectively. Other mathematical symbols are explained in Sec. 5 and summarized in the table in SI Sec. S1.

Added/Modified in the manuscript:

(In Sec. 5.1) The used mathematical symbols are summarized in the table in SI Sec. S1.

Regarding the red bars in the q plot, we believe you are referencing the visual representation in the Var-CRI illustration. Essentially, within the plots, the red bars depict a categorical distribution where the length of each red bar represents the probability of that particular realization. We clarified this point in the revised manuscript where the red bars appear the first time, as follows:

Added/Modified in the manuscript:

(In the caption of Fig. 1 and Fig. 2) The red bars depict a categorical distribution where the length represents the probability of a particular realization.

REVIEWER COMMENTS

Reviewer #1 (Remarks to the Author):

Remarks to the Author

Title: Collective Relational Inference for learning physics-consistent heterogeneous particle interactions

Recommendation: Accept with minor revisions

The authors' response has addressed my concerns. One minor suggestion is to check the case of the conferences and journals in the references, including [15][17][21].

Reviewer #2 (Remarks to the Author):

RESPONSE

Thanks for the careful, detailed reply as well as the extensive update to the paper. Your effort in making a good paper is great to see and the main reason why I slightly lean towards acceptance despite our disagreements.

Following your reply, I believe I now understand why we disagree on multiple points, and it's mostly a single reason. I believe NRI with fast modular meta-learning also collectively infers interactions: its encoder `_neural network_` is the same as the Kipf NRI, but its encoder really is the neural network embedded within the simulated annealing. This allows that paper to also collectively infer edges: each sampling within simulated annealing is independent, but the update function within the inner optimization is coupled because it simulates the system with all the edges.

Note that this is not just a philosophical definition of what the encoder is. If exactly one of the edges incoming a node has to be of 'green type', Kipf NRI can't deal with it and this paper can deal with it thanks to the neural architecture, but ModularMeta NRI also can

thanks to the iterative inference. This is because simulated annealing will keep resampling any invalid configuration.

- Therefore, I agree with the collective edge inference being novel with respect to Kipf NRI, but not w.r.t. the overall NRI literature. Of course, the way this coupling is implemented in this paper is different and novel to my understanding.

- Since the modular meta-learning encoder is different, I disagree with that being a good reason to not include them in the results. This is more the case given how much emphasis there is on collective inference being a key novelty.

- I understand the trouble of having to adapt a complex codebase to a new setup being only a single PhD student. Therefore, I'm not asking to run ModularMeta on new experiments. However, if we know that there is a baseline that runs on the exact same setup, the ethical/proper science thing is to include it. Therefore, I feel quite strongly that we should include ModularMeta wherever we already have the numbers.

-

OPINION AND SUGGESTIONS

- The emphasis on collective being a key novelty need to be toned down and rephrased as collective inference neural network/architecture as the main novelty. Furthermore, since ModularMeta is slow (bc of inner inference), this innovation is still valuable. This together with the temporal variability and the new experiments are still a good contribution.

- I think the ethical thing is to not hide ModularMeta results wherever the original paper provided them. I feel less strongly on having to run new experiments, as that requires effort.

- The sentences saying that ModularMeta having the same encoder as NRI should probably be substituted for encoder neural network and then add a comment that explains this point about collective inference through optimisation.

- If you include ModularMeta wherever you have the results you don't need "an excuse sentence" to not include it wherever you don't include it, readers understand researchers have finite time. Alternatively, you can criticise that it's much slower due to precisely this

inner inference instead of a feed-forward architecture.

- I appreciated the new experiments and improved writing, which go in line with my original observation of this paper being thorough and well-implemented.

I recommend acceptance, but strongly conditional on my suggestions: I believe the contribution is more subtle than the one currently being claimed via the text and the experiments, but the contribution is there and the paper is well executed.

Reviewer #3 (Remarks to the Author):

Thank you for your detailed responses and revisions to the manuscript. The robustness of the method to noise (with >20 deviations in position) was very impressive. The connection with symbolic regression is great. Given the additional examples and applications of the method and overall clarifications, I believe the work is now suitable for the broad audience of nature communications. There will likely still be hurdles to overcome in applying to real data, but the paper is stronger in terms of breadth.

I have a few additional suggestions:

1. Can you give some idea of the practical wall time required to run the method? On what sort of computer architecture? (apologies if this was provided somewhere, I did not see it).
2. While it may not be feasible to run a full search of the cut-off radius, it would be useful to demonstrate whether the algorithm is sensitive to this value for a least one of the examples.

Response to Reviewers' comments

We thank the reviewers for their helpful comments on our work. In response, we have made adjustments to the manuscript. We would like to highlight that we have conducted an additional experiment in which we evaluate CRI on the original dataset of NRI to provide a more convincing comparison between the proposed CRI and previous methods. In addition, we showcase how the cutoff radius influences the performance of Evolving-CRI.

In the following, we address the reviewer's comments in detail.

Reviewer 1

Summary The authors' response has addressed my concerns. One minor suggestion is to check the case of the conferences and journals in the references, including [15][17][21].

Reply: We thank the reviewer for pointing us to these typos. We updated the references.

Reviewer 2

Summary

- Following your reply, I believe I now understand why we disagree on multiple points, and it's mostly a single reason. I believe NRI with fast modular meta-learning also collectively infers interactions: its encoder neural network is the same as the Kipf NRI, but its encoder really is the neural network embedded within the simulated annealing. This allows that paper to also collectively infer edges: each sampling within simulated annealing is independent, but the update function within the inner optimization is coupled because it simulates the system with all the edges.
Note that this is not just a philosophical definition of what the encoder is. If exactly one of the edges incoming a node has to be of 'green type', Kipf NRI can't deal with it and this paper can deal with it thanks to the neural architecture, but ModularMeta NRI also can thanks to the iterative inference. This is because simulated annealing will keep resampling any invalid configuration.
- Therefore, I agree with the collective edge inference being novel with respect to Kipf NRI, but not w.r.t. the overall NRI literature. Of course, the way this coupling is implemented in this paper is different and novel to my understanding.
- Since the modular meta-learning encoder is different, I disagree with that being a good reason to not include them in the results. This is more the case given how much emphasis there is on collective inference being a key novelty.

- I understand the trouble of having to adapt a complex codebase to a new setup being only a single PhD student. Therefore, I'm not asking to run ModularMeta on new experiments. However, if we know that there is a baseline that runs on the exact same setup, the ethical/proper science thing is to include it. Therefore, I feel quite strongly that we should include ModularMeta wherever we already have the numbers.

Reply:

- We very much appreciate the detailed explanation. This helped us to better understand your point concerning the ModularMeta method and agree with you that it aims to capture the correlation among interactions on different edges through the simulated annealing optimization process. Therefore, we revised the manuscript (the details are described in the response to your suggestions below) to correctly and precisely describe the ModularMeta method and to be precise about our contributions and also about the novelty of the proposed approach.
- While ModularMeta and NRI-MPM are designed to capture the correlation between different edges, empirical results from a range of experiments indicates that these models fall short of achieving optimal efficacy in collective relational inference. Furthermore, it is crucial to underscore that the methodology employed by the proposed *collective relational inference* (CRI) framework is fundamentally distinct from the approaches taken by ModularMeta and NRI-MPM. The CRI framework is built upon a different set of assumptions and employs an alternative algorithmic structure that allows it to perform collective relational inference in a manner that is not only distinct but potentially more aligned with the intricate nature of relational networks.
- We agree that a comparison with ModularMeta would be helpful for the reader and we appreciate that you see the difficulty of applying ModularMeta to a new case study. Therefore, we reported the performance of CRI on the original simulation of NRI, to provide a convincing comparison between CRI and ModularMeta on the same setup. We provide more details about this additional comparison below, in response to your suggestion.

Opinion and suggestions

- The emphasis on collective being a key novelty need to be toned down and rephrased as collective inference neural network/architecture as the main novelty. Furthermore, since ModularMeta is slow (bc of inner inference), this innovation is still valuable. This together with the temporal variability and the new experiments are still a good contribution.

Reply: We agree that previous methods have aimed to capture the correlation between different edges. Therefore, we made sure that the revised manuscript does not state anywhere that we invent "collective inference" and that in every statement we are precise in describing the novelty of our work. For example, we modified the statement about the difference between CRI and previous methods in the Abstract, as follows:

Added/Modified in the manuscript:

(In Abstract) ... which possesses two distinctive characteristics compared to existing methods. First, it infers the interaction types of different edges *collectively* by explicitly encoding the correlation among incoming interactions with a joint distribution, ...

In Sec.2, we make a clear statement that we do not invent collective inference and explain the difference from previous works, as follows:

Added/Modified in the manuscript:

(In the beginning of Sec.2) We note that CRI is not the first work aiming to perform collective relational inference. As introduced in Sec. 1, Alet et al. [14] aim to implicitly capture the correlation among interactions through simulated annealing optimization, while Chen et al. [12] seek to capture this correlation through their proposed relational interaction neural network. However, CRI fundamentally differs in its approach to capturing the correlations among different edges.

Added/Modified in the manuscript:

(In Sec.2.1) The desired property of collective inference is achieved by introducing a joint distribution of the types of correlated interactions, based on the Bayesian rules. This explicitly encodes the correlation among different incoming interactions. Such a characteristic distinctly sets CRI apart from previous methods of relational inference as introduced in [7, 12, 18, 14].

We also adjusted the statement in Conclusion, as follows:

Added/Modified in the manuscript:

(In Sec.4) We proposed a novel probabilistic method for relational inference that operates collectively and demonstrated its unique ability to collectively infer heterogeneous interactions. This approach distinguishes itself from and addresses the limitations of previous methods of relational inference.

We are grateful to the reviewer for the cautionary note regarding claims of novelty and value the acknowledgment of CRI's temporal variability and time efficiency, as well as the recognition of the new experiments conducted.

- I think the ethical thing is to not hide ModularMeta results wherever the original paper provided them. I feel less strongly on having to run new experiments, as that requires effort.
Reply: We concur with the reviewer's suggestion and have accordingly conducted a comparison of CRI with ModularMeta using the original NRI dataset. The results, which demonstrate CRI's superior performance over all baseline methods on the Charge data set, is now reported in SI Sec. S5. In the case of the springs dataset, all methods under consideration reached a performance plateau at 99.9%, indicating that there is negligible scope for further improvement in this context. The summarized results are presented in the following table, as reported in Supplementary Information, Section S5:

Performance of CRI on the original dataset used by NRI, for the comparison with baselines. (Table S3 in the revised SI)

Model	Springs	Charged
NRI (Kipf et al. [8])	99.9 ± 0.0	82.1 ± 0.6
MPM (Chen et al. [9])	99.9 ± 0.0	93.3 ± 0.5
ModularMeta (Alet et al. [10])	99.9	88.4
CRI	99.9 ± 0.0	98.5 ± 0.4

- The sentences saying that ModularMeta having the same encoder as NRI should probably be substituted for encoder neural network and then add a comment that explains this point about collective inference through optimisation.

Reply: We thank the reviewer for this suggestion. We modified the sentences about ModularMeta in the Introduction, as follows:

Added/Modified in the manuscript:

(In Introduction) Therefore, Alet et al. [14] used the same encoder as NRI [7] to initially infer a proposal distribution for the interaction type of each edge. Subsequently, they used the simulated annealing optimization algorithm to sample possible configurations of the interaction type across different edges. The correlation among interactions on different edges is implicitly captured through this optimization process.

- If you include ModularMeta wherever you have the results you don't need "an excuse sentence" to not include it wherever you don't include it, readers understand researchers have finite time. Alternatively, you can criticise that it's much slower due to precisely this inner inference instead of a feed-forward architecture.

Reply: Thank you for your suggestion. In response, we have clarified our statement to specify that we excluded ModularMeta for our new dataset due to its slow performance. Additionally, as previously mentioned, we have now included a comparison of the two methods on the original NRI dataset. The revised manuscript is updated to reflect these changes as follows:

Added/Modified in the manuscript:

(In Sec. 3.2) Finally, we note that ModularMeta [14] is not included as a baseline in this benchmark due to technical reasons, in particular its slow performance owing to the inner inference. Nevertheless, we have evaluated CRI on the original dataset provided by Kipf et al.[7], where CRI demonstrated superior performance on the Charged dataset and matched the performance on the Springs dataset with a 99.9% accuracy, as detailed in SI Sec. S5.

- I appreciated the new experiments and improved writing, which go in line with my original observation of this paper being thorough and well-implemented.

Reply: We are grateful to the reviewer for their recognition and appreciation of our efforts.

Reviewer 3

Thank you for your detailed responses and revisions to the manuscript. The robustness of the method to noise (with >20 deviations in position) was very impressive. The connection with symbolic regression is great. Given the additional examples and applications of the method and overall clarifications, I believe the work is now suitable for the broad audience of nature communications. There will likely still be hurdles to overcome in applying to real data, but the paper is stronger in terms of breadth.

I have a few additional suggestions:

1. Can you give some idea of the practical wall time required to run the method? On what sort of computer architecture? (apologies if this was provided somewhere, I did not see it).

Reply: We run experiments on a server with RTX 3090 GPUs. Each experiment is run on a single GPU. We do not implement multi-GPU parallelism. This hardware information together with Python dependencies is provided in the associated code repository: <https://gitlab.ethz.ch/cmbm-public/toolboxes/cri> and in Sec. 5.6 of the revised manuscript.

It takes 9 hours for CRI to reproduce the original simulation dataset provided in NRI reference, which is the experiment reported in SI Sec. S5. Each VAR dataset in Sec. 3.1 takes 3 hours and the Netsim dataset takes 2.5 hours (while the graph size of Netsim is larger than that of VAR, Netsim needs less training epochs). The time needed for the particle simulations in Sec. 3.2 varies from 1 hour for five particles with 100 training examples to 47 hours for ten particles with 10000 training examples. The experiment of Evolving-CRI in Sec. 3.3 takes 30 hours.

We now provide this run-time information in Sec.5.6.

2. While it may not be feasible to run a full search of the cut-off radius, it would be useful to demonstrate whether the algorithm is sensitive to this value for a least one of the examples.

Reply: Thank you for this suggestion. In the crystallization simulation (Sec.3.3), particles interact with nearby counterparts within a specified cutoff radius. To streamline the input for the ML methods, we set each particle to interact with its five closest neighbors. In an additional case study, we varied the number of incoming interactions, which is equivalent to varying the cutoff radius. We report there results of this new experiment in the revised SI Sec. S15. the results are illustrated in the new Fig. S3, as follows:

The figure showing the accuracy of Evolving-CRI in terms of the cutoff radius. (Fig. S3 in the revised SI)

Our results indicate that the accuracy improves as we increase the cutoff radius (equivalently, when increasing the number_of_neighbors from 2 to 5), up to the point where it reaches the ground-truth cutoff radius (where number_of_neighbors equals 5).

REVIEWERS' COMMENTS

Reviewer #2 (Remarks to the Author):

Sorry for the late reply. All replies and adjustments work for me and I now agree with the message, claims, and content of the paper. I want to thank the authors for always delivering very high quality analysis, detailed responses and appropriate adjustments to the discussion.

Congrats also on the great paper, I hope it is accepted!

Reviewer #3 (Remarks to the Author):

Thank you to the authors for their careful responses to the issues raised by the other reviewers and me.

Reviewer #3 (Remarks on code availability):

I checked to see if the code was available and if there was a readme, but did not run the code.